# Bicyclic azetidines target acute and chronic stages of *Toxoplasma gondii* by inhibiting parasite phenylalanyl t-RNA synthetase

Joshua B. Radke [ORCID] [1], Bruno Melillo [ORCID] [2,3], Payal Mittal[4,5], Manmohan Sharma[4], Amit Sharma[4,5], Yong Fu[1], Taher Uddin[1], Arthur Gonse[3], Eamon Comer[3], Stuart L. Schreiber[3,6], Anil K. Gupta[2,7], Arnab K. Chatterjee[2,7] & L. David Sibley [ORCID] [1✉]

*Toxoplasma gondii* commonly infects humans and while most infections are controlled by the immune response, currently approved drugs are not capable of clearing chronic infection in humans. Hence, approximately one third of the world's human population is at risk of reactivation, potentially leading to severe sequelae. To identify new candidates for treating chronic infection, we investigated a series of compounds derived from diversity-oriented synthesis. Bicyclic azetidines are potent low nanomolar inhibitors of phenylalanine tRNA synthetase (PheRS) in *T. gondii*, with excellent selectivity. Biochemical and genetic studies validate PheRS as the primary target of bicyclic azetidines in *T. gondii*, providing a structural basis for rational design of improved analogs. Favorable pharmacokinetic properties of a lead compound provide excellent protection from acute infection and partial protection from chronic infection in an immunocompromised mouse model of toxoplasmosis. Collectively, PheRS inhibitors of the bicyclic azetidine series offer promise for treatment of chronic toxoplasmosis.

[1] Department of Molecular Microbiology, Washington University Sch. Med., St Louis, MO 63110, USA. [2] Department of Chemistry, The Scripps Research Institute, La Jolla, CA 92037, USA. [3] Chemical Biology and Therapeutics Science Program, Broad Institute, Cambridge, MA 02142, USA. [4] Molecular Medicine-Structural Parasitology Group, International Centre for Genetic Engineering and Biotechnology, New Delhi 110067, India. [5] National Institute of Malaria Research, New Delhi 110077, India. [6] Department of Chemistry and Chemical Biology, Harvard University, Cambridge, MA 02138, USA. [7] Calibr at Scripps Research, La Jolla, CA 92037, USA. ✉email: sibley@wustl.edu

*T*oxoplasma gondii is a widespread parasite of animals that causes zoonotic infections in people, infecting up to a third of the world's human population[1]. Acute infections caused by the actively proliferating tachyzoite stage are generally well controlled by the immune system[2,3]. In response, the parasite differentiates to a chronic stage called the bradyzoite that resides within tissue cysts that form primarily in muscle and brain[4,5]. The residence of these semi-dormant stages in sites of immune privilege like the brain, combined with their propensity to reactivate in response to declining systemic immunity, poses great risk due to re-emergence of actively proliferating tachyzoites that destroy tissue[6]. Additionally, healthy adults are at risk of recurrent ocular toxoplasmosis that can result in loss of vision due to infections that are more severe and difficult to treat in some regions, such as South America[7]. The current treatment for toxoplasmosis is based on inhibition of the folate pathway that is required for nucleotide synthesis in the parasite[8]. Such a strategy targets actively-replicating tachyzoites but spares bradyzoites, which reside within cysts in the host's muscle and brain tissue[8] and divide infrequently and asynchronously[9]. In order to successfully target the chronic tissue stages, compounds would thus need to be effective in crossing the blood-brain barrier, accumulating in the CNS, and target an essential process in the semi-dormant bradyzoite stage of the life cycle. This challenge is complicated by the relative lack of knowledge of what pathways are essential in bradyzoites[10].

A recent high-throughput screen (HTS) for small molecules that would act synergistically with interferon-gamma (IFN-γ)[11] identified several potent hits from the diversity-oriented synthesis (DOS) small-molecule library developed at the Broad Institute[12,13]. The DOS library, enriched in compounds typically underrepresented in commercial screening libraries, has proven valuable in identifying novel-mechanism-of-action probes and drug leads[14]. Bicyclic azetidines derived from DOS libraries have recently been shown effective in single-dose efficacy against multiple stages of *Plasmodium* spp., including both rapidly replicating asexual blood stages and slow-replicating or dormant hypnozoite stages[15]. Bicyclic azetidines have also been shown to be highly potent inhibitors of *Cryptosporidium* spp.[16,17]. In both cases, efficacy has been demonstrated in vitro and using mouse models of parasitic infection. Bicyclic azetidines have been shown to target apicomplexan phenylalanine tRNA synthase (PheRS)[15,16,18]. This essential enzyme that differs sufficiently from its human counterpart, making it an excellent target for the development of new therapeutics.

In the present study, we analyzed a series of bicyclic azetidines for their ability to inhibit *T. gondii* growth in vitro as tachyzoites, to target bradyzoites from tissue cysts produced in vivo, and to protect against acute and chronic toxoplasmosis in a relevant mouse model. Several compounds in this series exhibited high potency and selectivity for inhibition of *T. gondii* PheRS and provided effective inhibition of parasite growth in vitro and in vivo. These findings suggest that PheRS inhibitors based on the bicyclic azetidine scaffold deserve further attention as multistage inhibitors of *T. gondii*.

## Results

**Bicyclic azetidines are potent inhibitors of *T. gondii* tachyzoite growth.** In light of prior studies demonstrating the efficacy of PheRS inhibitors against apicomplexan parasites[15,16,18], we assessed the activity of 28 structurally unique compounds from the bicyclic azetidine series to determine their activities against the type II ME49 strain expressing firefly luciferase (FLuc)[19] tachyzoites of *T. gondii* using a luciferase-based growth screen to determine half-maximal effective concentration (EC$_{50}$) (Fig. 1a, Table S1). All compounds had EC$_{50}$ values lower than 5 μM,

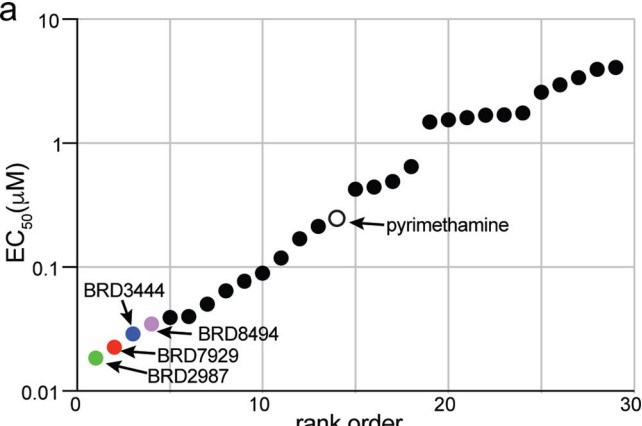

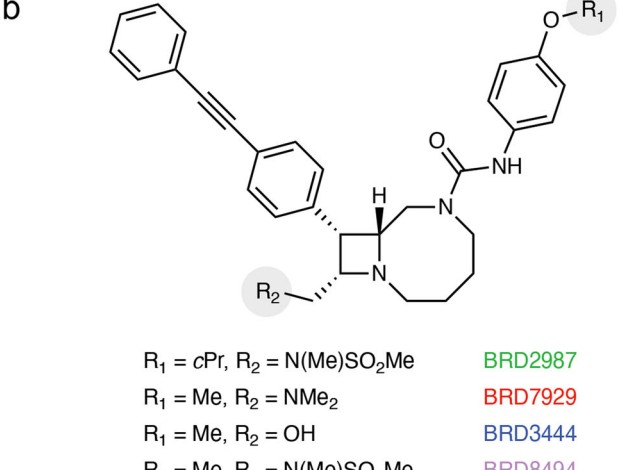

**Fig. 1 Bicyclic azetidines are potent inhibitors of *T. gondii* tachyzoite growth. a** Rank order based on median effective concentration (EC$_{50}$) determination of compounds from the bicyclic azetidine series against growth of *T. gondii* tachyzoites in vitro. Values represent average of two biological replicates used to calculate the EC$_{50}$ value with 10-point dose–response curves. The four most potent compounds are highlighted: BRD2987 (0.0195 μM, green); BR7929 (0.0226 μM, red); BRD3444 (0.0289 μM, blue) and BRD8494 (0.0348 μM, purple). Pyrimethamine (0.248 μM, open circle) highlighted for reference. A stepwise description of chemical synthesis scheme for BRD2987 is found in Fig. S1. A complete list of EC$_{50}$ values for all 28 bicyclic azetidines screened is found in Table S1. **b** Chemical structures of most potent bicyclic azetidines: BRD2987, BRD7929, BRD3444, and BRD8494, which differ only in the nature of appendages (R$_1$, R$_2$) as indicated. Source data are provided as a Source Data file.

including thirteen compounds with greater potency than pyrimethamine (EC$_{50}$ = 0.248 μM), which is part of the current standard of care (Fig. 1a). Among this series, ten compounds had sub 0.1 μM EC$_{50}$ values for *T. gondii* growth inhibition (Table S1). Four lead compounds were selected for further evaluation based on potency against the tachyzoite stage (Fig. 1a): BRD2987 (green, EC$_{50}$ = 0.0185 μM), BRD7929 (red, EC$_{50}$ = 0.023 μM), BRD3444 (blue, EC$_{50}$ = 0.0289 μM) and BRD8494 (purple, EC$_{50}$ = 0.0348 μM), differing in chemical substituents on the aryl urea (R$_1$) and azetidine ring (R$_2$, Fig. 1b). To test whether bicyclic azetidines were potent against multiple genetic lineages of *T. gondii*, we engineered FLuc expression into eight independent genetic backgrounds[20] (Table S2). BRD7929 was highly potent

**Table 1 Determination of EC$_{50}$ values of BRD7929 against firefly luciferase expressing parasites from a diverse cross section of *T. gondii* genotypes.**

| Strain | Type | Origin | EC$_{50}$ (nM) | SD |
|---|---|---|---|---|
| GT1 | I | N. America | 16.07 | 7.78 |
| RH | I | Europe | 24.14 | 3.93 |
| ME49 | II | Europe | 49.42 | 9.29 |
| CTG | III | N. America | 31.53 | 8.20 |
| MAS | IV | Europe | 41.97 | 15.61 |
| RUB | V | S. America | 26.89 | 9.55 |
| FOU | VI | Europe | 45.13 | 12.22 |
| VAND | X | S. America | 27.94 | 6.66 |

against all lineages with EC$_{50}$ values less than 50 nM against tachyzoite growth in vitro (Table 1).

**Bicyclic azetidines display similar activity against apicomplexan parasites.** To further understand the mechanism of action of bicyclic azetidines against *T. gondii*, we examined the correlation between EC$_{50}$ values for *T. gondii* tachyzoites with EC$_{50}$ values of blood-stage *P. falciparum* for which corresponding data was available[15]. We found a strong positive correlation for twenty compounds including the most potent inhibitors of *T. gondii* tachyzoites and blood-stage *P. falciparum* growth (Fig. 2a). Furthermore, we analyzed the stereochemical specificity of BRD3444 by comparing the EC$_{50}$ values from *T. gondii* and *P. falciparum* of this compound to those of its seven stereoisomers (Fig. 2b). The choice of BRD3444 in these assays was driven largely by the fact that we had all eight stereoisomers available for testing in parallel and there were existing data for *P. falciparum*. The highest activity against both parasites is associated with the *S, R, R* (C$_2$, C$_3$, C$_4$) and *R, R, R* isomers (Fig. 2b), indicating that the shared stereochemical configuration is critical to potency. The strong, positive correlation within the bicyclic azetidine scaffold and the nearly identical stereochemical selectivity for BRD3444 against *T. gondii* and *P. falciparum* imply a shared molecular target and mechanism of action. Expanding on this observation, we next correlated the EC$_{50}$ values of six bicyclic azetidines previously screened against *C. parvum*[16] with *T. gondii* values (Fig. 2a). Indeed, a strong correlation of activity was identified (Fig. 2a) between these parasites, including BRD3444 which was one of the most potent inhibitors of *T. gondii* growth (Fig. 1a) and which reduced oocyst shedding by 96% in a *C. parvum* mouse model of infection[16]. To broaden the potential therapeutic use of the series against other apicomplexans and infer common mechanisms of action, we determined the EC$_{50}$ values of eight bicyclic azetidines against the veterinary pathogen *Neospora caninum* (Fig. S2) with a 10-dose–response curve based on a transgenic *N. caninum* parasite expressing lacZ[21] (Table S3). Although only a small set of compounds were tested against *N. caninum*, they showed similar potencies to their activities against *T. gondii* as shown by linear regression ($r^2 = 0.69$) (Fig. S2). Similar to *T. gondii*, BRD2987 and BRD7929 were among the most potent compounds against *N. caninum* (Fig. S2, Table S3). Taken together, the strong positive correlation of potency of the bicyclic azetidine series against multiple species further supports a shared mechanism of action for bicyclic azetidines against apicomplexan parasites.

**Bicyclic azetidines inhibit *T. gondii* phenylalanine tRNA synthetase (TgPheRS).** Previous studies of *Plasmodium* and *Cryptosporidium* parasites identified cytoplasmic tRNA synthetase (PheRS) as the parasite target of the bicyclic azetidine chemical series[15,16]. The PheRS enzyme is highly conserved among

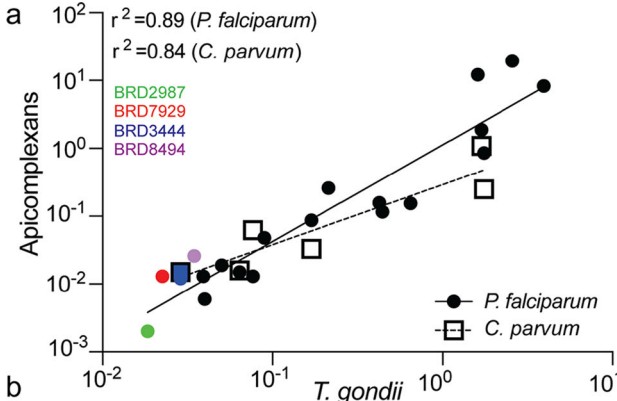

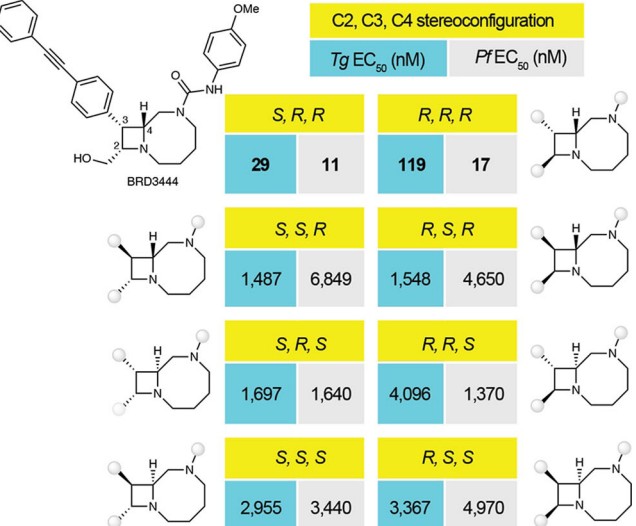

**Fig. 2 Bicyclic azetidines inhibit the growth of multiple species of Apicomplexa. a** Correlation of growth inhibition for *T. gondii* (Tg) tachyzoites compared with blood-stage *P. falciparum* (Pf)[15] (black regression line, $r^2 = 0.89$, $P < 0.0001$) based on EC$_{50}$ values for different bicyclic azetidines. A similar correlative analysis between *T. gondii* and *C. parvum* (Cp) (Tg vs Cp, open squares) EC$_{50}$ values for bicyclic azetidines (dashed regression line, $r^2 = 0.84$, $P = 0.0102$). Colored circles (Tg vs Pf) or squares (Tg vs Cp) highlight four lead compounds as identified in Fig. 1: BRD2987 (green); BR7929 (red); BRD3444 (blue) and BRD8494 (purple). See Table S2 for all EC$_{50}$ values against *C. parvum*[16]. **b** Stereospecificity of antiparasitic activity of bicyclic azetidine BRD3444. Chemical structures and in vitro antiparasitic activity of bicyclic azetidine BRD3444 and its seven stereoisomers; repeated appendages are depicted as gray beads for clarity, and the stereochemical configuration of C$_2$, C$_3$, C$_4$ is indicated (yellow boxes). Half-maximal effective concentrations of in vitro inhibition of *T. gondii* growth are shown (EC$_{50}$, green boxes, calculated as the mean of two biological replicates ± SEM.); previously reported EC$_{50}$ values of in vitro inhibition of *P. falciparum* growth[15] are indicated for comparison (gray boxes). Source data are provided as a Source Data file.

apicomplexans and mutations that confer resistance to bicyclic azetidines are conserved between *P. falciparum*[15] and *C. parvum*[16]. In order to identify the target of bicyclic azetidines in *T. gondii*, we selected for resistance in vitro using the compound BRD7929 due to favorable PK properties[15] that made it suitable also for in vivo infection studies described below. We evolved resistance to BRD7929 in two parallel, independent pools of parasites that were selected by sequential passages over the course of ~5 months (Fig. 3a). We determined the EC$_{50}$ value of each pool using a luciferase-based growth assay to monitor BRD7929 sensitivity

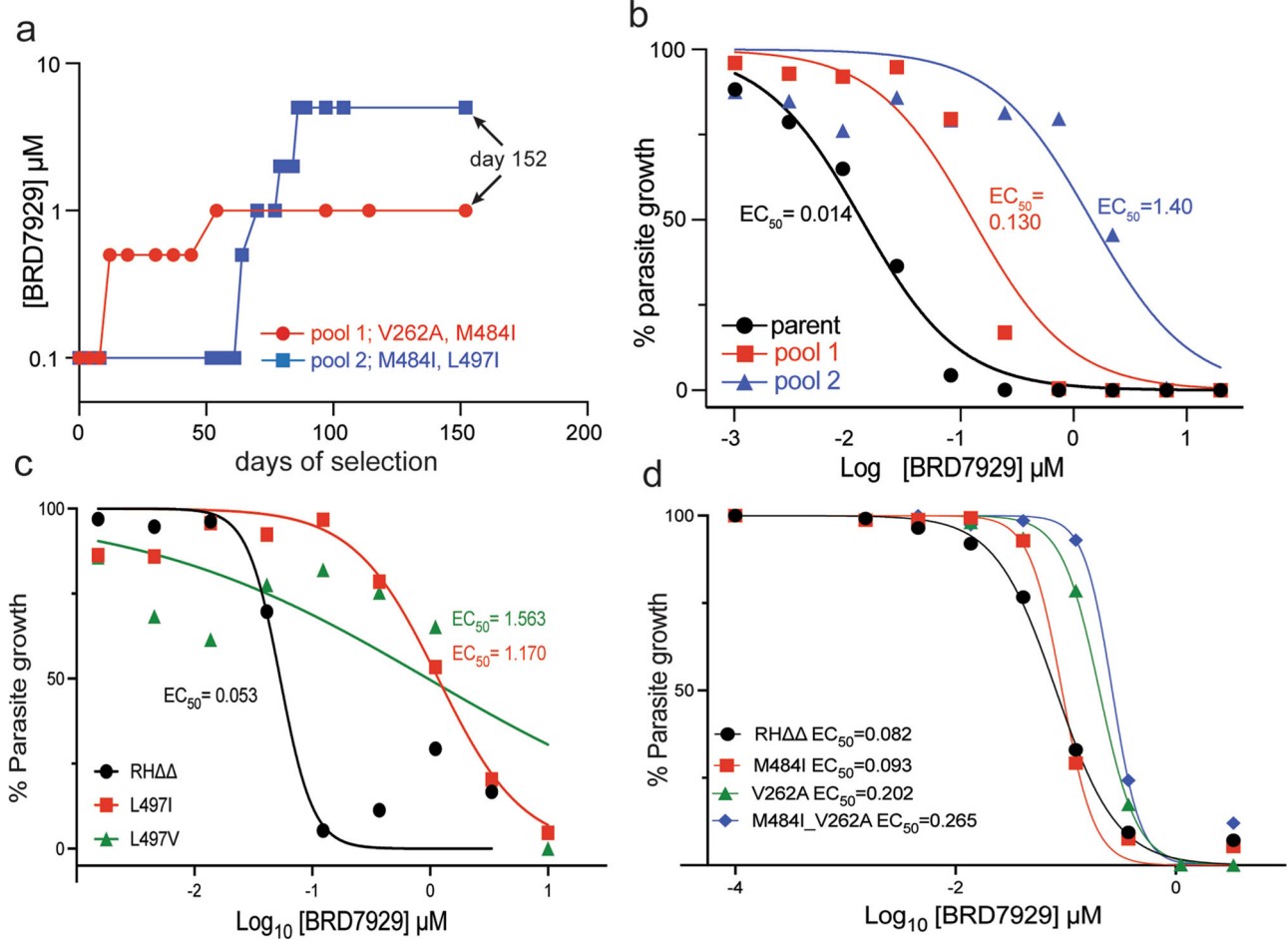

**Fig. 3 Resistance to BRD7929 correlates to resistance-conferring mutations identified in *P. falciparum*. a** Evolved resistance to BRD7929 in *T. gondii* (RH-FLuc strain) by serial passage at sub-lethal concentrations. Two independent pools of BRD7929 resistant parasites were selected over 152 days of continuous culture in increasing concentrations of BRD7929. Pool 1 = 1.0 μM final concentration, 12 passages, red line and circles; Pool 2 = 5.0 μM final concentration, 18 passages, blue line and boxes. Stepwise increase in BRD7929 started at 0.1 μM for both pools. Genomic DNA for whole-genome sequencing (WGS) was collected for analysis from each pool at day 152. **b** $EC_{50}$ determination of BRD7929 resistant pools using a 10-point dose–response curve. Parent (RH-FLuc) = black circles and line; pool 1 = red squares and line; pool 2 = blue squares and line. **c** Evaluation of $EC_{50}$ for *T. gondii* tachyzoites engineered to carry TgPheRS resistance-conferring mutations TgPheRS[L497V] and TgPheRS [L497I]. **d** Evaluation of $EC_{50}$ for *T. gondii* tachyzoites engineered to carry TgPheRS resistance-conferring mutations TgPheRS[M484I] TgPheRS[V262A] and the double mutant. Mutant lines are shown color; the wild-type RHΔΔ parent is shown in black. All $EC_{50}$ values are presented as the mean of three biological replicates ± SEM (n = 3). Source data are provided as a Source Data file.

relative to the parental line (Fig. 3b). Pool 1 ($EC_{50} = 0.130$ μM, red line) and pool 2 ($EC_{50} = 1.40$ μM, blue line) showed resistance to BRD7929 ranging from 10- to 100- fold, respectively (Fig. 3b). Subsequent whole-genome sequencing identified three non-synonymous single nucleotide variants (SNV) associated with BRD7929 resistance in the gene encoding the alpha subunit of cytosolic PheRS (i.e. TGGT1_2345050, ToxoDB.org). The SNVs encoded new amino acid changes at positions TgPheRS[L497I] (pool 1) TgPheRS[V262A] (pool 2), and TgPheRS[M484I] (pools 1 and 2) in TgPheRS (Fig. 3a). The mutation TgPheRS[L497I] occurred in the residue corresponding to the previously described resistance mutation PfPheRS[L550V] in *P. falciparum*[15], while the other mutations were unique to *T. gondii* (Fig. S3). To determine if these amino acid changes conferred resistance to BRD7929, we used a CRISPR/Cas9-based markerless editing strategy to introduce point mutations into a wild-type *T. gondii* background (type I, RH strain lacking *hxgprt* and *ku80*[22]) and evaluated growth using a modified parasite lytic assay[23]. Mutation of TgPheRS[497] from a leucine to either a valine to mimic the *P. falciparum* mutation, or isoleucine

as seen in *T. gondii* (Fig. 3c), resulted in 40–60-fold changes in $EC_{50}$ values over the parent strain, indicating that these single point mutations indeed confer resistance to BRD7929. Single point mutations of methionine to isoleucine (TgPheRS[M484I]) or valine to alanine at position 262 (TgPheRS[V262A]), only shifted the $EC_{50}$ value 2 or 4-fold, respectively (Fig. 3d). Sequential introduction of the TgPheRS[M484I] and TgPheRS[V262A] mutations to generate a double mutant only shifted the $EC_{50}$ slightly more than every single mutation (Fig. 3d), consistent with the lower overall resistance of pool 1. We did not detect any difference in the fitness of the mutant lines based on plaque formation or lytic growth on fibroblast (HFF) monolayers (Fig. S4). Collectively, the identification and re-introduction of mutations in TgPheRS that confer tachyzoite growth in the presence of BRD7929 strongly suggest that PheRS is the primary target of the bicyclic azetidines in *T. gondii*.

**BRD7929 is a potent and selective inhibitor of TgPheRS.** To provide further support that TgPheRS is the molecular target of bicyclic azetidines, we assayed the compound against purified

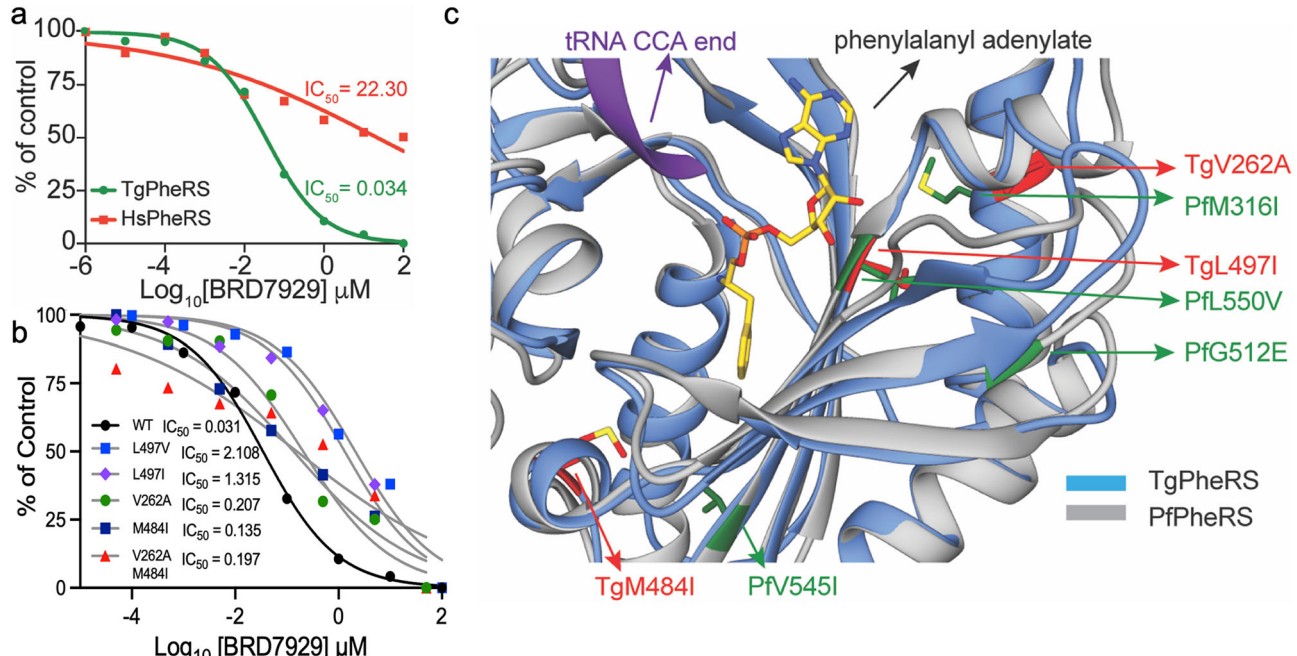

**Fig. 4 BRD7929 is both selective for and potent against TgPheRS. a** Determination of the median inhibition concentration (IC$_{50}$) of BRD7929 for *T. gondii* TgPheRS (green) versus human HsPheRS (red) using a 10-dose–response curve. **b** Inhibition curves for wild-type TgPheRS enzyme (black) vs. TgPheRS containing resistance-conferring point mutations. Average of three biological replicates each with two technical replicates ($n = 6$) ± SEM. **c** Homology models showing catalytic pocket of PfPheRS (gray) and TgPheRS (blue) in the α-subunit superimposed on the structure of TtPheRS-L-Phe-AMP-tRNA (yellow, PDB 2IY5). The drug-resistant PfPheRS and TgPheRS mutations are highlighted in green and red, respectively. Source data are provided as a Source Data file.

**Table 2 Evaluation of host cell toxicity for bicyclic azetidines.**

| ID | *T. gondii* EC$_{50}$ (μM) | HEPG2 EC$_{50}$ (μM) | HEPG2 fold selectivity | HFF EC$_{50}$ (μM) | HFF fold selectivity | THP-1 EC$_{50}$ (μM) | THP-1 fold selectivity | SH-SY5Y EC$_{50}$ (μM) | SH-SY5Y fold selectivity | A549 EC$_{50}$ (μM) | A549 fold selectivity | Caco-2 EC$_{50}$ (μM) | Caco-2 fold selectivity |
|---|---|---|---|---|---|---|---|---|---|---|---|---|---|
| BRD7929 | 0.023 | 3.065 | 133.3 | 7.872 | 342.3 | 0.672 | 29.2 | 0.731 | 31.8 | 5.374 | 233.7 | 7.003 | 304.5 |
| BRD3444 | 0.029 | 11.327 | 390.6 | 20[a] | 689.7 | 2.578 | 88.9 | 2.665 | 91.9 | 20[a] | 693.2 | 7.816 | 269.52 |
| BRD8494 | 0.035 | 20[a] | 571.4 | 20[a] | 571.4 | 0.461 | 13.2 | 1.785 | 51.0 | 20[a] | 571.4 | 20[a] | 571.4 |
| BRD2987 | 0.019 | 20[a] | 1052.6 | 20[a] | 1052.6 | 0.255 | 13.4 | 2.355 | 123.9 | 2.166 | 114.0 | 20[a] | 1052.6 |

EC$_{50}$ values based on two biological replicates each with two technical replicates. Values represent means.
[a]Curve fitting EC$_{50}$ values for these compounds were greater than the highest drug concentration (20 μM) used in the screen. 20 μM was used to determine fold selectivity. Fold selectivity was calculated as EC$_{50}$ host cell/EC$_{50}$ *T. gondii*.

recombinant proteins for human (HsPheRS) and wild-type TgPheRS (Fig. 4a). In these studies, we focused on BRD7929 due to its superior PK properties[15] that made it a better candidate for in vivo studies described below. First, we determined the half-maximal inhibitory concentration (IC$_{50}$) for BRD7929 against HsPheRS and TgPheRS to evaluate selectivity for the parasite enzyme over the host enzyme (Fig. 4a). BRD7929 inhibited TgPheRS in a concentration-dependent manner and was >600-fold selective for the parasite enzyme (Fig. 4a) Consistent with this finding, the most active bicyclic azetidines showed minimal toxicity for host cells and greater than 100-fold selectivity for inhibiting parasite growth over host cell lineages such as fibroblasts (HFF), lung endothelial cells (A549), kidney hepatoma cells (HepG2) and intestinal epithelium (Caco-2) (Table 2). Somewhat greater toxicity was seen using a human monocyte like tumor line (THP-1) or a neuroblastoma cell line (SH-SY5Y), although even with these more sensitive lines there was greater than 10-fold, and in most cases > 30-fold, selectivity for parasite growth inhibition *vs.* host (Table 2). Introduction of resistance-conferring

mutations at amino acid TgPheRS[L497V], which mimics the change seen in *P. falciparum*, shifted the IC$_{50}$ ~60-fold (Fig. 4b). Additionally, the introduction of the new point mutations seen in resistant populations of *T. gondii* into the wild-type enzyme shifted the IC$_{50}$ values (Fig. 4b), with the most dramatic changes resulted from the mutation of TgPheRS[L497I]. Much smaller differences were observed in the single mutants TgPheRS[M484I], TgPheRS[V262A] or the corresponding double mutant (Fig. 4b). Overall, there was an excellent correlation between the IC$_{50}$ values of enzyme inhibition and EC$_{50}$ values for parasite growth inhibition ($r^2 = 0.92$ (Table S4)). To illustrate amino acid changes associated with BRD7929 resistance, we modeled TgPheRS and PfPheRS onto the human PheRS (HsPheRS) structure (PDB 3L4G) and mapped the resistance-conferring mutations for each enzyme on their respective ribbon models (Fig. 4c). Among these mutations, only TgPheRS[L497I] and PfPheRS[L550V] are located within the enzyme active site, while others are close by and likely act allosterically (Fig. 4c). Taken together, the close relationship between genetic and biochemical

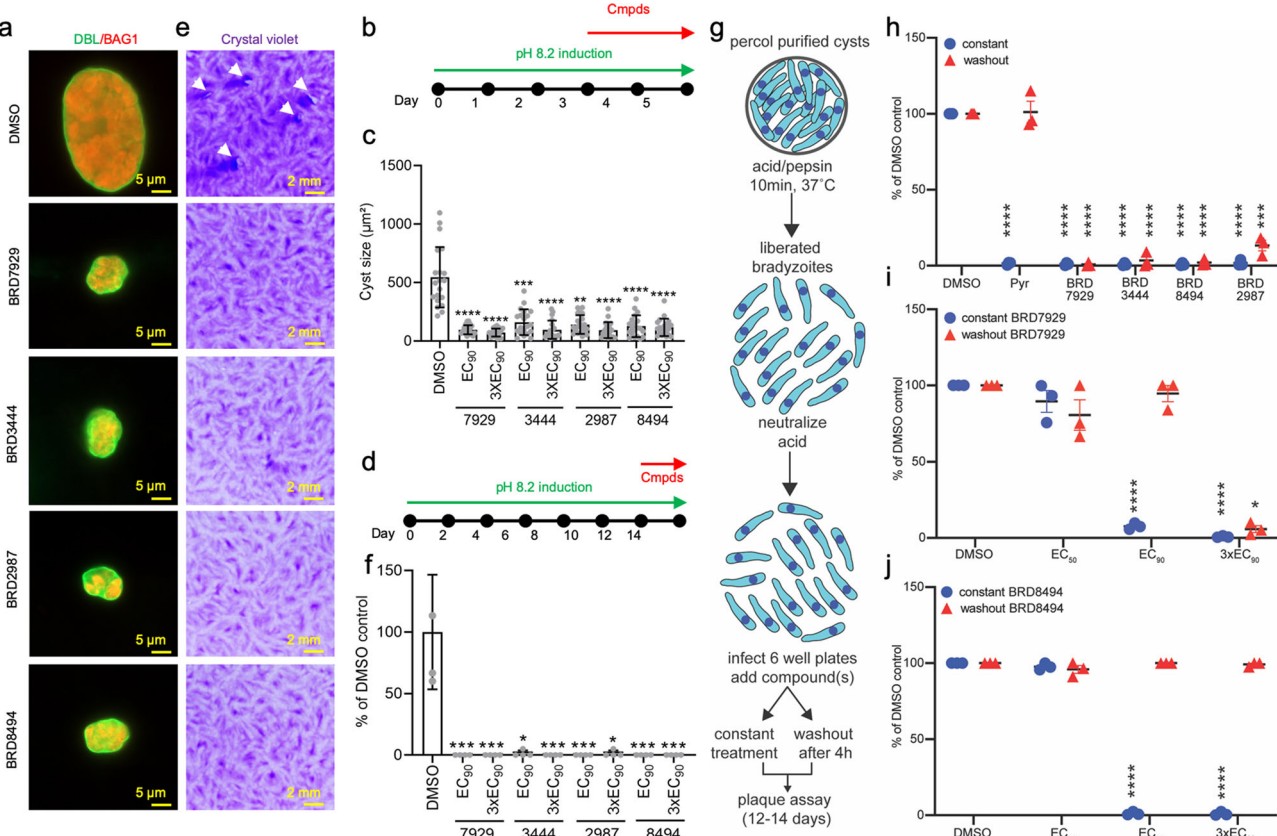

**Fig. 5 Lead compounds from bicyclic azetidine series inhibit growth of ex vivo derived bradyzoites. a** Immunofluorescence analysis (IFA) of effects of compounds on the development of in vitro induced cysts. Bradyzoites were induced at pH 8.2 for 3 days, followed by another 2 days treatment with compounds at $EC_{90}$ and $3xEC_{90}$. DMSO was used as vehicle control. DBL was used to detect the cyst wall (green) while BAG1 served as bradyzoite specific marker (red). Scale bars = 5 μm. **b** Timeline for monitoring the effect of compounds (Cmpds) on cyst size. **c** Cysts size was measured for individual cysts following compound treatment or DMSO control. Data are plotted as means ± SD ($n = 20$ cysts from 2 independent experiments) and analyzed by two-sided Kruskal–Wallis test with Dunn's multiple comparison correction test, $**P = 0.0005$, $***P = 0.0003$, $****P < 0.0001$. **d** Timeline for long term in vitro induction of bradyzoites followed by plaque assay. Cysts were induced by culture at pH 8.2 for 14 days including treatment with compounds for the final 48 h. **e** Plaque assay for the effects of compounds on the viability of bradyzoites induced for 14 days. Conditions as shown in (**d**). Scale bars = 2 mm. **f** Quantification of plaque numbers in (**e**). Data were normalized to the average of the DMSO treatment group and plotted as means ± SD for 3 technical replicates from a single experiment. Two-sided Kruskal–Wallis test with Dunn's multiple comparison correction test, $*P = 0.0154$, $***P = 0.007$. **g** Schematic representation of Percoll purification of tissue cysts harvested from chronically infected mice and subsequent treatment regimen. Percoll purified tissue cysts of ME49-FLuc strain (25 cysts/6 well plate) are treated with acid/pepsin to liberate encysted bradyzoites (10 min, 37 °C) followed by neutralization with sodium carbonate solution. Liberated bradyzoites are aliquoted into 6 well plates and are either under constant or 4 h washout treatment conditions and evaluated by plaque assay on day 12–14. **h** Ex vivo derived bradyzoites were subjected to constant (blue circles) or 4 h washout (red triangles) at 2.5 μM for each compound. DMSO (vehicle) and pyrimethamine (2.5 μM, growth) were included as controls. Mixed effects analysis with Dunnett's multiple comparisons test, $****$, $P < 0.0001$, $***$, $P = 0.0003$. **i** Treatment of liberated bradyzoites with BRD7929 at $EC_{50}$: = 0.023 μM, $EC_{90}$: = 0.098 μM, $3xEC_{90}$: 0.293 μM as indicated (blue circles = constant; red triangles = 4 h washout). Two-way ANOVA with Šidák's multiple comparisons test $*$, $P = 0.0299$; $****$, $P < 0.0001$, **j** Inhibition of ex vivo derived bradyzoites by BRD8494 in constant (blue circles) or 4 h washout (red triangles) treatment conditions at three concentrations: $EC_{50}$: 0.035 μM; $EC_{90}$: 0.095 μM; and BRD8494. $EC_{50}$, $EC_{90}$ and $3xEC_{90}$ values determined using 10-dose–response curve described in Fig. 1. Two-way ANOVA with Šidák's multiple comparisons test, $****$, $P < 0.0001$. Data represent the mean of three independent biological replicates each with ≥ 2 technical replicates ($n \geq 6$). Solid black line equals the mean and red intervals represent ± SEM. All statistically significant changes are based on comparison to DMSO control for each treatment group. Source data are provided as a Source Data file.

resistance in *T. gondii* further corroborates TgPheRS as the molecular target of BRD7929.

**Lead compounds from bicyclic azetidine series inhibit growth of in vitro differentiated and ex vivo derived bradyzoites.** To determine if bicyclic azetidines could target and eliminate bradyzoites, we developed several assays to monitor their effects against in vitro bradyzoites using alkaline pH to induce differentiation[24]. Initially, we induced bradyzoites during a 3 day period of culture at pH 8.2 in ambient $CO_2$ during which they

convert to stages that express BAG1 and develop a cyst wall that stains with *Dolichos biflorus* lectin (DBL) (Fig. 5a). Following 3 days induction, compounds were added for an additional 2 days and then the size of cysts was measured to evaluate continued growth (Fig. 5b). Treatment with the top 4 compounds (BRD7929, BRD3444, BRD2987, and BRD8494) at the $EC_{90}$ or $3xEC_{90}$ significantly diminished cyst size (Fig. 5c). To further examine the sensitivity of in vitro derived bradyzoites to bicyclic azetidines, we extended the in vitro differentiation to 14 days and treated in vitro derived cysts for the last 48 h of induction (Fig. 5d). Following treatment, monolayers were disrupted and

**Table 3 Pharmacokinetic parameters of BRD7929 after oral administration to mouse at repeated dosing of 10 mg/kg per day with analysis after day 1, 4, and 8.**

| Parameters[a] | Day 1 | Day 4 | Day 8 |
|---|---|---|---|
| $C_{max}$ (ng/mL) | 153 | 330 | 207 |
| $T_{max}$ (h) | 18.3 | 4.3 | 5.0 |
| $AUC_{0-24}$ (h·ng/mL) | 3,187 | 6357 | 4390 |
| $AUC_{0-inf}$ (h·ng/mL) | NA | 27,667 | 19,950 |
| $t_{1/2}$ (h) | NA | 64.6 | 81.3 |
| MRT (h) | 13.1 | 11.7 | 12 |
| CL/F (mL/min/kg) | NA | 7.1 | 10 |
| $V_d$ (L/kg) | NA | 32 | 62 |
| $K_p$ ($C_b/C_p$) | 169 | 211 | 577 |
| $K_{pu,u}$ ($C_b/C_p$) | 2.1 | 2.6 | 7.2 |

[a]The value represents the mean.

the cysts were purified using DBL conjugated to beads followed by trypsin to liberate the bradyzoites. Purified bradyzoites were then cultured on monolayers of HFF cells for 14 days to allow the formation of plaques in the absence of compounds. Treatment with the top 4 compounds significantly reduced plaque formation (Fig. 5e,f), indicating that they were effective at reducing the viability of late stage in vitro derived bradyzoites.

Because in vitro derived bradyzoites do not represent the fully differentiated forms that occur during chronic infection in vivo[25,26], we also developed an assay to monitor the effects of compounds on ex vivo bradyzoites isolated from tissue cysts from chronically infected mice[9,27]. Cysts were purified from the brains of chronically infected mice using Percol gradients, followed by disruption with pepsin/HCl, neutralization, and treatment in vitro followed by monitoring growth by plaque assay (Fig. 5g). Treatment with the top 4 compounds for either 4 h followed by washout or under constant treatment conditions almost completely eliminated growth when used at high concentrations (i.e. 2.5 μM) (Fig. 5h). To further explore the sensitivity of ex vivo bradyzoites to treatment, we assayed BRD7929 (Fig. 5i) and BRD8494 (Fig. 5j) at $EC_{50}$, $EC_{90}$, and $3 \times EC_{90}$ concentrations based on tachyzoite growth assays (Table S1). At the $EC_{50}$, BRD7929 only partially reduced growth compared to vehicle controls (DMSO) (Fig. 5i), indicating that bradyzoites are slightly less susceptible than tachyzoites. Increasing BRD7929 to the $EC_{90}$ or $3 \times EC_{90}$ resulted in 93% and 99% growth inhibition during constant treatment and this effect was sustained following washout after short-term treatment at $3 \times EC_{90}$ (Fig. 5i). Constant treatment with BRD8494 did not inhibit the growth of liberated bradyzoites at the $EC_{50}$, yet both higher concentrations nearly eliminated growth under constant treatment (Fig. 5j). Treatment followed by washout for BRD8494 failed to inhibit growth regardless of concentration (Fig. 5j). Collectively, the dose-dependent activity of bicyclic azetidines against in vitro differentiated and ex vivo derived bradyzoites demonstrate that these compounds are capable of acting on both acute and chronic stages of T. gondii.

**Pharmacokinetics studies.** Previous pharmacokinetics (PK) studies of BRD7929 in mouse found that it has good oral bioavailability (80%) and a very stable half-life in plasma following a single administration[15]. Therefore, we chose to focus on this compound for further in vivo studies. In order to examine the PK properties over a longer time frame, we started with conditions that had previously been used (i.e. 10 mg/kg) for single-dose studies and performed a repeated dosing study by administering BRD7929 at 10 mg/kg p.o. once a day for eight days with interval

sampling during days 1, 4, and 8. Consistent with previous findings, the compound reached steady state levels in plasma during the first 24 h with modest $C_{max}$ and AUC values (Table 3) and then remained relatively constant during the 8 day period, albeit with a minor increase during day 4 (Fig. 6a). Compound BRD7929 is highly hydrophobic and not surprising, it showed high binding to plasma proteins (96.01%) and brain tissues (99.95%). In order to estimate the levels of free compound in plasma and brain we plotted the concentrations of unbound compound vs. time and compared them to values for growth inhibition obtained in vitro. Comparison of these curves indicates that the concentrations achieved in plasma were well below the $EC_{50}$ values (Fig. 6a). Because toxoplasmosis is a major concern for CNS infection, we were also interested in the ability of BRD7929 to reach the brain and accumulate there over time. Sampling of brain tissue indicated that BRD7929 was readily absorbed in brain tissue and in fact accumulated in tissue relative to plasma, likely due to a high volume of distribution (Vd) (Table 3). Comparison of the unbound concentrations in brain vs. plasma (i.e. $K_{p\ u,u}$) indicated that BRD7929 is concentrated ~ 2–7 fold in brain vs. plasma from days 1 to 8 (Table 3), indicating that it has a favorable profile for CNS indications. Plotting the free concentrations achieved in plasma indicated that the compound remained below the $EC_{50}$ for most of the duration (Fig. 6a). In contrast, the free levels of compound in the brain exceeded the $EC_{50}$ and nearly reached the $EC_{90}$ by day 8 (Fig. 6a). We also investigated higher doses of oral administration of BRD7929 but found that increasing the dose to 30 mg/kg did not lead to a proportional increase in plasma levels and instead resulted to overt signs of toxicity. Although the basis for this adverse effect was not investigated further, we decided to adopt the dose of 10 mg/kg for efficacy studies.

**Compound BRD7929 protects immunocompromised mice from acute and chronic infection.** To further evaluate the potency of bicyclic azetidines for multistage inhibition of T. gondii growth, we utilized interferon-gamma receptor knockout mice (Ifngr1$^{-/-}$), which are completely susceptible to T. gondii infection because they cannot mount an effective immune response[28]. To determine whether the bicyclic azetidine BRD7929 could protect from T. gondii oral challenge, we infected Ifngr1$^{-/-}$ mice with freshly isolated tissue cysts (type II, ME49-FLuc strain) by oral gavage and allowed 48 h post-challenge before starting three parallel treatment regimens (Fig. 6b). Mice were treated once a day (QD) by oral gavage with BRD7929 for 10 days or 20 days (Fig. 6c) at 10 mg/kg. We compared these treatments to sulfadiazine (sulfa, 0.25 g/L in drinking water)[29], which prevents death from acute infection, but allows the development of bradyzoite containing tissue cysts. Treatment with BRD7929 for both 10 days (mean survival = 16.5 days) and 20 days (mean survival = 21 days) greatly extended the survival time vs. sulfadiazine alone (mean survival = 10 days) (Fig. 6c). Moreover, treatment with BRD7929 prevented death in 3 of 8 mice when the treatment was extended to 20 days (Fig. 6c), indicating a longer treatment window is significantly more effective in preventing the establishment of latency and/or reactivation. Importantly, surviving mice were seropositive indicating they had become infected (Fig. S5). As these animals are profoundly immunocompromised, their survival for this length of time after completion of treatment suggested that their infections were cured. To confirm that surviving animals were not harboring infectious parasites, they were sacrificed, the brain removed, homogenized, and gavaged into recipient naïve Ifngr1$^{-/-}$ mice that were left untreated and monitored for 20 days. All recipient mice survived and remained seronegative (Fig. S5), confirming that treatment had resulted in

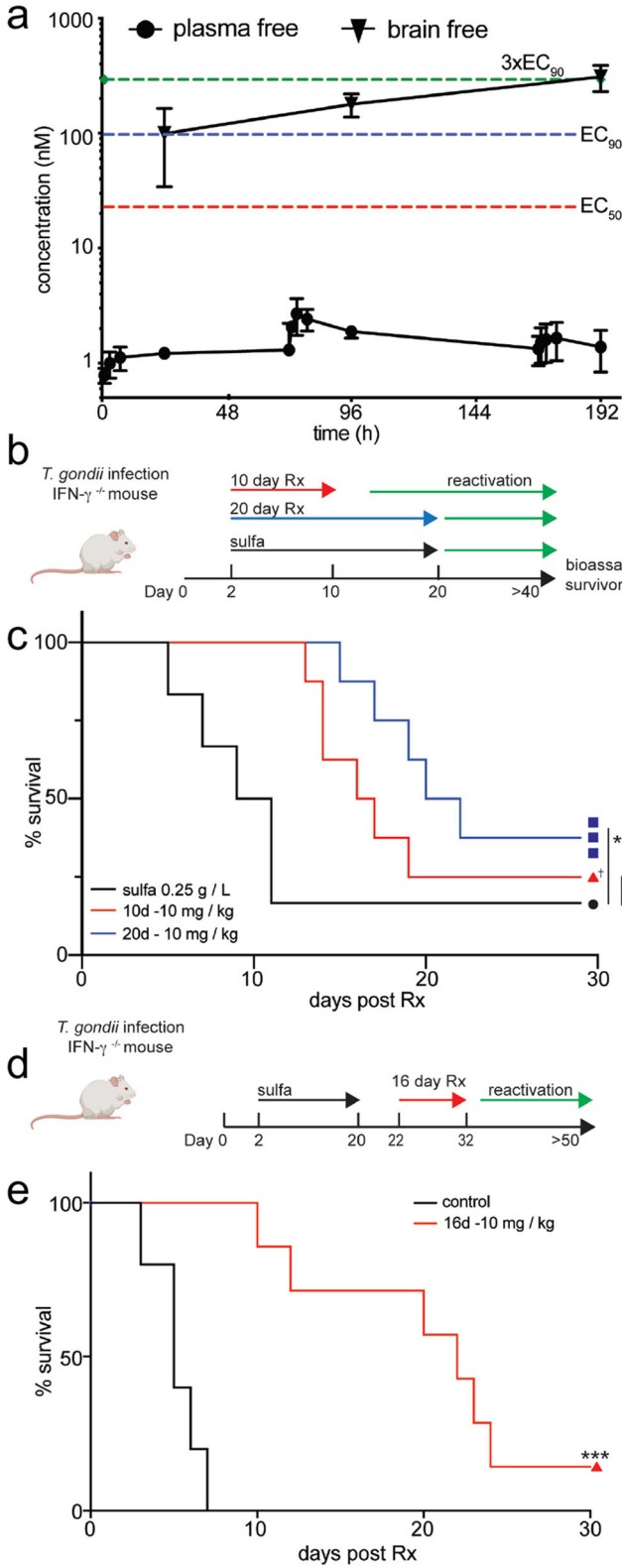

**Fig. 6 BRD7929 accumulates in the brain and protects Ifngr1$^{-/-}$ mice from oral challenge with tissue cysts and prevents establishment of chronic infection. a** Repeated dose PK studies of BRD7929 in mouse. Compounds were administered daily p.o. at 10 mg/kg for 8 days with sampling at 1, 3, 7, and 24 h on day 1, 4, and 8 ($n = 3$ animals per group, repeated twice, with one experiment plotted). Values plotted are means ± S.D. were based on unbound fractions in plasma or brain. Values for $EC_{50}$, $EC_{90}$, and 3 x $EC_{90}$ were based on in vitro studies with tachyzoites. **b** Schematic of oral challenge protection model for treatment of toxoplasmosis. Ifngr1$^{-/-}$ mice were infected with the ME49-FLuc strain. Two days later mice were divided into three treatment cohorts: (1) sulfadiazine (sulfa, 0.25 g/L in drinking water, $n = 6$ (3 male, 3 female), black arrow) for 20 days, (2) BRD7929 at 10 mg/kg p.o. once per day for 10 days (red arrow, $n = 8$ (4 male, 4 female)) or (3) BRD7929 at 10 mg/kg p.o. once per day for 20 days (blue arrow, $n = 8$ (4 male, 4 female)). Mice were then monitored for reactivation (green arrow) by survival and weight loss. **c** Surviving mice were monitored for at least 29 days post-treatment. Mice treated for 20 days with BD7929 survived significantly longer than sulfadiazine mice (*, $P = 0.0487$, Mantel-Cox test). † = two surviving mice from the 10-day cohort reached day 29, however, one succumbed to infection on day 33. All surviving mice were positive for *T. gondii* infection by ELISA and verified to be cured by bioassay into naïve recipient mice (Fig. S5). **d** Diagram of reactivation model for chronic toxoplasmosis including treatment regimen. Ifngr1$^{-/-}$ mice were infected with the ME49-FLuc strain by oral gavage. Animals were placed on sulfadiazine from day 2–20 (0.25 g/L in drinking water). Two days after removal of sulfadiazine animals were treated with vehicle (black, $n = 5$ (2 male, 3 female)) or once a day with BRD7929 at 10 mg/kg p.o. for 16 days (red line, $n = 7$ (4 male, 3 female)). Reactivation of chronic infection (green arrow) was monitored for at least 20 days post-treatment. **e** Control mice (vehicle, black) quickly succumb to recrudescent infection within 10 days. Mice treated with BRD7929 survived significantly longer, including one animal that was verified to be cured by bioassay into naïve recipient mice (***, $P = 0.0003$, Mantel-Cox test). Images of mice created with BioRender. Source data are provided as a Source Data file.

tested whether treatment could prevent reactivation of latent infection using the Ifngr1$^{-/-}$ mouse model of chronic infection (Fig. 6d)[27]. We infected Ifngr1$^{-/-}$ mice with tissue cysts by oral gavage (type II, ME49-FLuc) and treated with sulfadiazine for 20 days starting 2 days post-infection to allow for the establishment of chronic infection and prevent death by acute infection. Because sulfadiazine has no activity against tissue cysts, discontinuation of treatment results in reactivation of chronic infection and re-emergence of tachyzoite proliferation leading to death in 7–10 days[28]. Following removal of sulfadiazine (2 days), animals were treated QD by oral gavage for 16 days with BRD7929 (10 mg/kg) or vehicle control (Fig. 6d). Without sulfadiazine, control mice rapidly succumbed to recrudescent infection, whereas animals receiving BRD7929 had a delayed onset of death, including 1 of 6 animals who survived >20 days post-treatment window (Fig. 6e). This sole surviving animal was seropositive (Fig. S5a), and subsequent bioassay of brain homogenate into a naive recipient Ifngr1$^{-/-}$ mouse failed to detect any residual infection. Collectively, these findings indicate that this surviving animal was cured of chronic infection by treatment with BRD7929.

cure in the donor animal. Collectively, these findings indicate that BRD7929 prevented death during acute infection and either prevented tissue cysts from forming or eliminated them during chronic infection. Overall, 4 of 16 animals were cured by treatment with BRD7929.

To further understand the potential of BRD7929 as a therapeutic for the treatment for chronic toxoplasmosis, we

## Discussion

Current therapies for toxoplasmosis are effective at controlling acute infection caused by proliferating tachyzoites while having minimal impact on chronic infection caused by semi-dormant bradyzoites[8]. To identify new inhibitors that might address this

critical need, we evaluated bicyclic azetidines that have previously been shown to be multistage inhibitors of apicomplexan parasites[15,16]. There was a strong correlation between the potency of bicyclic azetidines in inhibiting growth of *T. gondii*, *P. falciparum*, and *C. parvum*, suggesting a common mechanism of action. Comparison of the inhibitory activity of bicyclic azetidines against purified enzymes revealed they are potent and selective inhibitors of *T. gondii* PheRS relative to the human counterpart. Moreover, mutational studies, based both on directed point mutations and evolved resistance, identified PheRS as the primary target of bicyclic azetidines in *T. gondii*. The lead compound in this series BRD7929 demonstrated good bioavailability and concentration in the CNS and was able to control acute infection as well as partially prevent the establishment and reactivation of chronic infection in a relevant mouse model. Collectively, these studies indicate that PheRS is essential, druggable, and provides an excellent target for further studies to optimize inhibitors for control of acute and chronic toxoplasmosis. Additionally, unlike many other inhibitors that show more selective profiles[30], bicyclic azetidines have broad efficacy across the Apicomplexa as shown here for *T. gondii* and *N. caninum* and in previous studies on *Cryptosporidium*[16] and *Plasmodium*[15,18].

The current standard of care for toxoplasmosis is based on inhibition of the folate pathway using a combination of pyrimethamine with a sulfa drug (e.g. sulfadiazine or sulfamethoxazole) that act synergistically to block DNA/RNA synthesis and prevent replication[8]. Although effective in controlling acute infection, there are issues with allergic reactions to the sulfa component, bone marrow suppression that requires co-administration of leucovorin (folinic acid), and adverse reactions to pyrimethamine[31]. These issues are particularly problematic where long-term treatment is required to control disease such as in immunocompromised patients[8] or in cases of recurrent ocular infection, which is common in South America[32]. Additionally, this combination treatment, while synergistic in controlling tachyzoite proliferation, is unable to eradicate chronic infection, presumably because bradyzoites divide slowly and asynchronously and hence they are not susceptible to the inhibitory action of anti-folates[8]. Other antibiotics that also show efficacy, including tetracycline, clindamycin, azithromycin, and related analogs, also only act against the actively proliferating tachyzoite form[8]. In contrast, atovaquone[29,33] and quinolones[34,35], both of which inhibit the cytochrome bc1 complex of the mitochondria, show some efficacy in reducing the burden of tissue cysts in vivo. Unfortunately, atovaquone is limited by treatment failures[36–38] that may result from high-frequency emergence of drug resistance as shown in laboratory studies[39,40]. Although quinolones suffer from limited solubility that hampers bioavailability, this problem has been partially resolved using esterified prodrugs[35] or tetrahydroquinones, which have a similar molecular target[41]. Notably, neither of these compounds is able to completely eradicate chronic infection, similar to other recently described compounds that show efficacy in the mouse model of toxoplasmosis[42]. Other recent studies show that modulation of immune responses using the TLR7 agonist imiquimod can reduce chronic infection and suppress subsequent infection by treated bradyzoites in an immunocompetent mouse model[43]. Using a more stringent model for reactivation of chronic infection, it was shown that ATP mimetics that specifically inhibit calcium-dependent protein kinase 1 (CPKD1) both reduce the burden of chronic infection[44] and prevent reactivation in an immunocompromised mouse model of toxoplasmosis[27,45]. Importantly, in this chronic model of reactivation, compounds must both control tachyzoite growth and eliminate all residual cysts to allow survival of the animal, thus providing a rigorous endpoint to monitor radical cure.

Apicomplexan parasites contain three genes encoding phenylalanine tRNA synthase that are compartmentalized in the mitochondria, apicoplast, and cytosol where they independently participate in protein translation[46]. Previous studies have shown that bicyclic azetidines are potent and specific inhibitors of the alpha subunit of the cytosolic PheRS from *P. falciparum*[15] and *C. parvum*[16]. Our current studies extend these findings to demonstrate that bicyclic azetidines are also potent and selective inhibitors of cytosolic *T. gondii* PheRS. The core azetidine ring in the bicyclic azetidines contains three chiral centers, resulting in eight topologically distinct stereoisomers that exhibit very different geometries. Comparison of these eight stereoisomers revealed that the stereochemical specificity of *T. gondii* growth inhibition by bicyclic azetidines was highly similar to that reported previously for *P. falciparum*[15]. These findings suggest that the binding mode of bicyclic azetidines is highly similar among these two species, although it is evidently much different from HsPheRS resulting in >500-fold selectivity. Homology modeling of the PfPheRS and TgPheRS enzymes predicts a high degree of similarity that allowed us to compare sites that resulted in resistance to bicyclic azetidines. In this regard, position 497 in TgPheRS, which corresponds to position 550 in PfPheRS, is highly informative as the mutant TgPheRS[L497V], similar to that seen in resistant *P. falciparum*[15], renders *T. gondii* highly resistant to inhibition by bicyclic azetidines like BRD7929. A similar mutation of CpPheRS[L482V] in *C. parvum* results in resistance to bicyclic azetidines, emphasizing the conserved nature of this binding interaction in compound potency[16]. Additionally, selection of resistance mutations in *T. gondii* resulted in a similar change in TgPheRS[L497I], which also resulted in high-level resistance. Additional point mutations in TgPheRS that were identified by whole-genome sequencing of resistant pools of parasites resulted in much lower levels of resistance when introduced as single mutations or in combination in wild-type parasites. Our studies were performed in a type I strain due to available genetic tools that facilitate the introduction of point mutations needed to validate the function of individual residues. However, we expect that similar observations would be made using other strain types as there are no polymorphisms in the PheRS gene among commonly used lab strains and the potency of compounds in inhibiting strains from widely different lineages was highly similar. Despite the fact that resistance mutations were readily obtained by a prolonged passage in vitro, their occurrence in vivo is likely to be less common given that parasite burdens rarely reach high levels during chronic infection and there is the limited human-to-human transmission.

Our findings demonstrate that bicyclic azetidines are potent against tachyzoites at low nM levels and treatment of bradyzoites in vitro at $EC_{90}$ or $3xEC_{90}$ concentrations results in almost complete killing, consistent with the potent and irreversible effects of these inhibitors against other apicomplexans[15–17]. Although it is clear that bicyclic azetidines targeting PheRS provide broad-spectrum inhibition of apicomplexan growth, the conservation of this target also complicates the goal of designing compounds specific to each parasite. Further structural and medicinal chemistry efforts may uncover specificity in the binding pocket that could be exploited for selectivity. Alternatively, it may be possible to take advantage of the tissue compartmentalization to drive specificity. Compounds that are effective against the asexual stages of *P. falciparum* would likely benefit from prolonged plasma exposure[15]. In contrast, studies of inhibitors of PI(4)K in *C. parvum* have emphasized the importance of lower bioavailability to maintain intestinal luminal concentrations without raising the risk of systemic exposure[47], although PheRS inhibitors like BRD7929, which has good oral bioavailability, are also effective in this model[16]. Finally, since the target organ for chronic *T. gondii* infection is the brain and muscle, compounds with CNS penetration and high tissue distribution may be used to preferentially target this organism.

Compound BRD7929 exhibits several highly desirable properties as a lead compound for treatment of toxoplasmosis. It has good oral bioavailability, a long half-life, high volume of tissue distribution, good potency, and a selective profile. These properties combined to provide excellent protection from acute infection and partial protection from chronic reactivation in an immunocompromised mouse model of toxoplasmosis. However, there are several limitations with BRD7929 that currently limit its utility for further in vivo studies. High protein binding results in low exposure of free compound and combined with the high volume of tissue distribution may contribute to lack of increased plasma exposure with higher dosing in vivo. In preliminary trials, increased repeated dosing with BRD7929 also resulted in adverse symptoms, which may be related to its previously described effects on ion channels[15]. Finally, free concentrations of BRD7929 in the brain tissue exceeded the $EC_{50}$ value but did not quite reach the $EC_{90}$ level, which likely resulted in the observed partial protection from chronic infection. In this regard, the somewhat higher toxicity for bicyclic azetidines for monocytic and neuronal lineages highlights an additional liability of the current lead. Further medicinal chemistry efforts to increase tissue exposure without such adverse effects will be needed to fully explore the pharmacodynamic properties and to identify more potent leads for eradicating chronic T. gondii infection. Provided improved leads can be identified, it would be important to extend the findings reported here to test challenge with different life cycle stages such as oocysts, to monitor the efficacy of treatment on development and shedding of sexual stages, and to examine other disease models including ocular and congenital infection.

## Methods

**Small-molecule synthesis**. Molecules employed in this study were prepared using procedures derived from previously published methods[15,16]. Specific synthetic schemes, procedures, and analytical data are available in the Supporting Information (Fig. S1).

**Animals**. Ifngr1$^{-/-}$ (B6.129S7-Ifngr1 $^{tm1Agt}$ /J) and CBA/J mice were purchased from Jackson Laboratory and bred locally at Washington University. Animal studies were conducted according to the U.S. Public Health Service policy on human care and use of laboratory animals. Animals were maintained in facilities approved by the Association for Assessment and Accreditation of Laboratory Animal Care. Studies were approved by Division of Comparative Medicine, Washington University. Sex and age matched mice between 8 and 12 weeks of age were used to perform experiments. Animals were maintained on a 12:12 light cycle, room temperature maintained at 70°F ± 2°F, and room humidity maintained at 50% ± 20%.

**Parasite strains and host cell culture**. *Toxoplasma gondii* strains RHΔhxgprtΔku80 (RHΔΔ)[22], RH-FLuc (type I)[48], and ME49-FLuc (type II)[19], were previously reported. New transgenic lines created here are found in Table S2. The *Neospora caninum* strain expressing β-galactosidase was described previously[21]. Tachyzoites from *T. gondii* and *N. caninum* were maintained by serial passage in confluent monolayers of human foreskin fibroblasts (HFF) as previously described[21,30]. HFF and parasite cultures were grown in 10% DMEM (DMEM supplemented with glutamine (10 mM), gentamycin (10 μg/mL, and 10% fetal bovine serum) incubated at 37 °C with 5% $CO_2$ and were verified mycoplasma free by the e-Myco Plus kit (Intron Biotechnology).

**Genome editing and parasite transfection**. In vitro cultures of HFF cells containing mature vacuoles were scraped, passaged through 23 g blunt end needles, and filtered using 3 micron filters. Purified parasites were resuspended in 350 μL of cytomix plus[49] and combined in a BTX 4 mm gap cuvette with 30 μL of PCR amplified templates corresponding to mutations of interest and 10 μg of pSAG1:Cas9-GFP U6:sgTgPheRS$^{[mut]}$ (in 20 μL, 400 μL total volume, see Supplementary Data 1 for sgRNA sequences) prior to electroporation using a BTX ETM 830 electroporator. Amplified templates for TgPheRS mutations were generated from gBlocks (Integrated DNA Technologies, Coralville, IA, USA) that contain a BclI restriction site that replaces the PAM sequence to protect the repair template from being cut by Cas9. At 16 h post-transfection, transgenic parasites expressing pSAG1:Cas9-GFP were sorted on a Sony SH800 FACS sorter directly into 96-well plates. Single clones containing TgPheRS$^{[mut]}$ were identified by amplifying a ~500 bp PCR fragment around introduced mutation followed by

digestion using BclI (New England Biolabs) BclI sensitive clones were expanded and the locus was Sanger sequenced to verify incorporation of each independent mutation. To generate firefly luciferase (FLuc) expressing strains, freshly harvested parasites were combined with 20 μg of pTUB-FLUC DHFR flox plasmid and 10 μg of pSAG1:Cas9-GFP U6:sgUPRT[50] (in 50 μL cytomix plus) and transformed by electroporation as described above. Parasites were allowed to recover overnight followed by sequential selection with pyrimethamine (1.0 μM) and 5-flurodeoxyuracil (FUDR, 10 μM), as described previously[48]. Stable clones were isolated by limiting dilution and a clone from each genetic background selected based on high luciferase activity (Table 1).

**Parasite growth and $EC_{50}$ determination assays**. All assays were carried out in a 96-well plate format containing confluent HFF monolayers to support parasite growth. To avoid edge effects, only the inner 60 wells of each plate were used. All compounds described were provided by the Broad Institute as 10 mM stocks in 100% DMSO and stored at −80 °C prior to use. Pyrimethamine (Sigma-Aldrich) was prepared as 5 mM stock in 100% DMSO and stored at −20 °C prior to use.

*Tachyzoite lytic assay*. Parasite growth on HFF monolayers was monitored using a 96-well plate-based lytic assay modified from a previously described protocol[23]. BRD7929 was diluted in 10% DMEM media to create a 20 μM solution that was then serially diluted 1:3 in DMEM across each 96-well plate to generate a 10-dose series of BRD7929 at 2X final concentration (in 100 μL vol). Freshly harvested parasites ($7.5 \times 10^4$ /well) were added to each well (in 100uL volume) to reach the final concentration range (10 μM to 0.0005 μM, 200 μL/well, 0.1% DMSO). Plates were incubated for 72 h at 37 °C then culture medium was aspirated and monolayers were fixed with 100% ethanol (ETOH) for 5 min at room temperature (RT). After rinsing with water, wells were stained with 0.1% crystal violet solution for 10 min at RT. Plates were rinsed with water, air dried, and absorption quantified on the Cytation 3 multi-mode imager at 570 nm wavelength. % parasite growth was calculated as 100−[(OD value for infected cultures at each concentration of compound/average OD value for no compound and no parasite infection) × 100]. $EC_{50}$ values were calculated based on the mean of three biological replicates with each replicate containing two technical replicates.

*Luciferase-based tachyzoite growth assay*. Luciferase assays were performed using the inner 60 wells of a clear bottom, white 96-well plate (Costar) to avoid well-to-well interference as previously detailed[11,30]. Compounds were diluted to generate a 3-fold dilution series as described above. Freshly harvested ME49-FLuc parasites ($5 \times 10^3$) were added to each well (in 100 μL volume) to yield 1x final compound concentration (200 μL/well total volume, 0.1% DMSO in 10% DMEM medium). Plates were incubated for 72 h at 37 °C prior to analysis using the Luciferase Assay System protocol (Promega) and described[11,30]. Briefly, culture media was replaced with 30 μL of 1x Cell Culture Lysis Reagent (Promega) and incubated for 10 min at RT. Following cell lysis, 100 μL of LAR reagent was added to each well and luciferase activity measured using a Beckman Coulter instrument. All liquid handling steps (compound serial dilution, media exchange and luciferase steps) were completed on a Dual Pod Biomek FX driven by the SAMI EX software system to ensure efficient and uniform execution of assays across all replicates (High-Throughput Screening Center, Washington University School of Medicine).

*Neospora caninum β-galactosidase assay*. β-galactosidase assays were conducted using tissue culture grade 96-well plates (TPP) containing confluent monolayers of HFF cells. A 10-point compound dilution series (2-fold) for each compound was generated as described above (2X concentration, 0.1% final DMSO). Growth of the *N. caninum* strain expressing lacZ was monitoring using a β-galactosidase (β-gal) assay as described in[21] with the following modifications. *N. caninum* lacZ expressing parasites ($1 \times 10^3$) were added (in 100 μL volume) into 96-well plates containing HFF monolayers to reach final compound concentration (5.0 μM to 0.002 μM in 200 μL total volume). Plates were incubated at 37 °C, 5% $CO_2$ for 72 h prior to completing β-gal assay as previously described[23,27,51]. Briefly, parasite containing monolayers were lysed with 1.0% Triton X-100 and β-gal activity determined by adding 1 mM chlorophenolred-β-D-galactopyranoside followed by monitoring absorption at 570 nM on a Cytation 3 multi-mode imager (BioTek), as described[51].

**Cloning of PheRS and generation of mutants**. For expression and purification of TgPheRS (*Toxoplasma gondii* cytoplasmic-Phenylalanine tRNA synthetase) enzyme, the amino acid sequence for the α (TGME49_234505) and β (TGME49_306960) subunits were retrieved from the ToxoDB (https://toxodb.org). The genes were synthesized and codon optimized by Geneart for expression in *E. coli* cells. The gene for α-subunit was cloned into the pETM11 and gene for β-subunit was cloned in pETM20 expression vectors employing Nco1 and Kpn1 restriction sites. Both the subunits (α and β) were co-transformed in *E. coli* BL-21 competent cells. Subsequent resistance conferring mutations were introduced using pETM11-TgPheRS plasmid using a site directed mutagenesis approach and assembled using the NEBuilder HiFi DNA Assembly kit according to manufacturer's protocol. All pETM11-TgPhers$^{[mut]}$ plasmids were Sanger sequenced to verify presence of mutation prior to use. See Table S3 for primer information.

Similarly, for HsPheRS (*Homo sapiens* cytoplasmic- Phenylalanine tRNA synthetase) enzyme, the full-length sequence for α (UniProtKB - Q9Y285 (SYFA_HUMAN)) and β (UniProtKB - Q9BR63 (Q9BR63_HUMAN)) subunits were retrieved using their uniport IDs. Both the subunits of the HsPheRS protein were synthesized by Geneart and co-transformed in *E. coli* B834 cells.

**Protein expression and purification**. For purification of both wild type and mutant TgPheRS, *E.coli* cultures were grown at 37 °C to an OD$_{600}$ of 0.6–0.8. recombinant protein expression was induced by the addition of 0.6 mM isopropyl β-d-1-thiogalactopyranoside (IPTG) at 18 °C. After 18–20 h post-induction, the cells were harvested by centrifugation at 5,000 g for 20 min, resuspended in binding buffer (50 mM Tris–HCl (pH 8), 200 mM NaCl, 4 mM β-mercaptoethanol (βMe), 10% (v/v) glycerol, 1 mM phenylmethylsulfonyl fluoride (PMSF) and 0.1 mg/mL lysozyme), and lysed using sonication. The lysed cells were then centrifuged at 20,000 g for 45 min and the cleared soluble protein lysate was loaded on a pre-packed Ni-NTA column (GE Healthcare). Proteins which bound the Ni-NTA were washed with 20 column volumes of binding buffer supplemented with 20 mM imidazole to remove impurities. Bound protein was eluted using a concentration gradient of imidazole from 0 to 1 M in elution buffer (80 mM NaCl, 50 mM Tris–HCl (pH 8), 4 mM β-mercaptoethanol, 10% (v/v) glycerol, 1 M imidazole, using AKTA-FPLC system (GE healthcare)). The fractions which contained purified protein were pooled together and concentrated using 30-kDa cut-off centrifugal devices (Millipore) followed by size exclusion chromatography using the GE HiLoad 60/600 Superdex column in a buffer containing 50 mM HEPES (pH 8), 200 mM NaCl, 4 mM β-mercaptoethanol, 1 mM MgCl$_2$. Purity of the eluted fractions was verified on a SDS PAGE gel, fractions pooled together and stored at −80 °C until further use.

**Enzyme assay**. The enzyme-inhibition assays for the recombinant wild-type TgPheRS, TgPheRS mutants, and HsPheRS proteins were performed as per earlier published reports[15,18,46]. Briefly, the assays were performed using malachite green based aminoacylation assay that determined the first step of the reaction by measuring the amount of phosphate released. All the enzymatic assays were performed in clear flat bottom 96-well plates at 37 °C with 100 nM recombinant PheRS enzymes in a total volume of 50 µL. The assays were done in a standard aminoacylation buffer comprising 50 mM MgCl$_2$, 150 mM NaCl, 30 mM HEPES (pH 7.5), 30 mM KCl, 30 mM DTT. The assays were initiated by adding 50 µM L-phenylalanine, 100 µM ATP and 2 U/mL *E. coli* inorganic pyrophosphatase (NEB). For determination of IC$_{50}$ values a 10-fold dilution of BRD7929 was done in aminoacylation assay buffer starting with 100 µM to final concentration of 0.000001 µM and the reaction mixture was incubated for 2 h at 37 °C. The reaction was stopped using malachite green stop solution and incubated at room temperature for 5 min. Enzyme activity was quantified using SpectraMax M2 (Molecular Devices) at 620 nm. MBP (maltose binding protein) was used a negative control for protein binding the reaction mixture. All the experiments were repeated three independent times with internal triplicates.

**TgPheRS homology modeling**. The TgPheRS (ToxoDB Gene ID: TGME49_234505) and PfPheRS (*Plasmodium falciparum* cytoplasmic- Phenylalanine aminoacyl tRNA synthetase) (PlasmoDB gene ID: PF3D7_0109800) homology model was built using Prime (Schrödinger Release 2015-2: Prime, version 4.0, Schrödinger)[52,53] with HsPheRS PDB 3L4G as a template. HsPheRS was chosen as template based on the identity and highest sequence similarity determined via PSI-BLAST. Cluster Omega was used to make the target-template alignment[54] using the EMBL server (https://www.ebi.ac.uk/Tools/msa/clustalo/) and the model was built and refined using the default settings in Prime. Figures for display were prepared with Chimera[55]. The TtPheRS-L-Phe-AMP-tRNA (PDB 2IY5 structure was used to mark the amino acid binding site (L-Phe), ATP site and tRNA binding site on the PfPheRS and TgPheRS homology models.

*Evolved resistance to BRD7929 in* T. gondii *tachyzoites and identification of resistance conferring mutations*. Tachyzoites of the strain RH-FLuc (clone B2)[48] were serially passaged at sub-lethal concentrations of BRD7929 (starting concentrations 0.1 µM (~3X EC$_{50}$). Concentrations of BRD7929 were increased step-wise (0.1 µM to 0.5 µM to 1.0 µM to 5.0 µM, respectively) when parasites underwent normal lytic growth on 48–72 h cycle for one to two weeks. Genomic DNA (gDNA) was isolated from resistant parasites from pool 1 (1.0 µM final concentration) and pool 2 (5 µM final concentration) using a DNeasy Blood and Tissue DNA isolation kit (Qiagen). Following gDNA purification, 1–3 µg of gDNA from RH-FLuc (parent), and each of the BRD7929 resistant pools was submitted to the Genome Technology Access Center at Washington University for whole genome sequencing (WGS) on the Illumina NovaSeq6000 platform. Sequencing analysis was completed using CLC Genomics Workbench (v20, Qiagen). Resequencing Analysis using Tracks function and mapping pool 1 and pool 2 genomes against the RH-FLuc parent to identify genome level mutations (using default parameters).

*In vitro tachyzoite growth assays*. Parasite growth on HFF monolayers was monitored by plaque formation using a 6 well plate assay. Freshly harvested parasites (200/well) were added to each well. Plates were incubated for 7 day at 37 °C in DMEM-10%FBS, then the culture medium was aspirated and monolayers were fixed with 100% ethanol for 5 min at room temperature. After rinsing with water, wells were stained with 0.1% crystal violet solution for 10 min at room temperature. Plates were rinsed with water, air dried, and plaques were quantified on a Zeiss AxioObserver microscope equipped with a 2.5x objective. Wells were imaged using a ChemiDoc Imager (BioRad) equipped with white light conversion screen and plaques area were measured using ImageJ software. Data were generated based on two biological replicates with each replicate containing three technical replicates.

Parasite growth on HFF monolayers was monitored using a lytic growth assay in 96-well plates. Freshly harvested parasites (2 × 10$^4$ /well) were added to each well in 2$^{nd}$ column and then serially diluted 1:2 in media to generate a 10-step dilution series. Plates were incubated in DMEM-10% FBS culture medium for 72 h at 37 °C then culture medium was aspirated, and monolayers were fixed with 100% ethanol for 5 min at room temperature. After rinsing with water, wells were stained with 0.1% crystal violet solution for 10 min at room temperature. Plates were rinsed with water, air dried, and absorption quantified on the Cytation 3 multi-mode imager at 570 nm. The growth curves were generated in Prism (GraphPad) based on the mean of two biological replicates with each replicate containing three technical replicates.

*PK studies*. BRD7929 was dosed orally at 10 mg/kg as a suspension in 10% ethanol, 4% Tween 80 and 86% saline to female CD1 mice with three mice per dosing group. All mice were fed prior to dosing. After dosing, 200 uL of blood was collected at predetermined time points (pre, 1, 3, 7, 24 h). Blood samples were processed for plasma by centrifuging at approximately 4 °C, 4,500 g for 15 min within min of collection. Plasma samples were stored in polypropylene tubes, quick frozen over dry ice and kept at −70 ± 10 °C until LC/MS/MS analysis. Plasma concentration versus time data was analyzed by non-compartmental approaches using the Phoenix WinNonlin 6.3 software program. Brain tissue samples were taken at day 1, 4, and 8. Tissue samples were processed by centrifugation at approximately 10 °C, 17,000 *g* for 30 min within 30 min of collection.

*In vitro bradyzoite growth assays*. To monitor the effects of compounds on early bradyzoite differentiation, ME49-Fluc tachyzoites were inoculated onto HFF monolayers grown on coverslips in 24-well plates and allowed infect during a 2 h incubation in 3% DMEM (DMEM containing glutamine (10 mM), gentamycin (10 µg/mL,) and 3% FBS) at 37°C, 5% CO$_2$. The medium was replaced with RPMI 1640 pH 8.2 and cells were cultured in ambient CO$_2$ at 37°C for 3 days to induce bradyzoites. Monolayers were treated with compounds at EC$_{90}$ or 3XEC$_{90}$ for another 2 days during culture in RPMI 1640 pH 8.2 at ambient CO$_2$ and 37°C. Samples were fixed and stained by biotinylated *Dolichos biflorus* lectin (DBL) (Vector Laboratories (#B-1035-5) 1,1/000) followed by Alexa Fluor 488 streptavidin (Thermo Fischer (#S11223) 1:1,000 dilution) and mouse mAb 8.25.8 anti-BAG1 (from Dr. Louis Wiess) (1:1,100)[56] followed by goat anti-mouse IgG conjugated to Alexa Fluor 568 (Thermo Fischer (#A-1104) 1:1,000). Samples were analyzed using a Zeiss AxioObserver Z1 equipped with Colibri LED illumination using a 40x EC Plan-Neofluar objective (N.A. 1.3) and images were captured using an ORCA-ER digital camera operated using ZEN v3.3 (Carl Zeiss). Cyst size was determined by measuring the area defined by the perimeter of the cyst wall as measured in microns.

To monitor the effects of compounds on more mature in vitro derived bradyzoites, ME49-Fluc tachyzoites were inoculated onto HFF monolayers grown in T25[2] flasks and bradyzoites were induced for 12 days in RPMI 1640 pH 8.2 at ambient CO$_2$ and 37°C. Cultures were then treated with compounds at EC$_{90}$ or 3XEC$_{90}$ for another 2 days. At the end of 14 days, monolayers were scraped and passed through a 23 g needle to liberate cysts, followed by centrifugation (400 × *g*, 10 min, 4 °C). Pellets were resuspended in Pearce$^{TM}$ streptavidin magnetic beads (Thermo Fisher) pre-coupled with biotinylated DBL (Vector laboratories) and incubated for 1 h at 4 °C. Beads were collected and washed twice using a magnetic stand, followed by treatment with 0.25 mg/ml trypsin for 10 min to liberate bradyzoites. Released bradyzoites were collected from the supernatant and counted using a hemocytometer. Bradyzoites were inoculated onto 6-well plates confluent with HFF monolayers and cultured for 14 days, followed by ethanol fixation and staining with 0.01% crystal violet. Plaques or foci of infection were imaged using an Axiovert 100 microscope equipped with an AxioCam color camera and analyzed by ImageJ.

*Ex vivo bradyzoite treatment assays*. The brains from CBA/J mice chronically infected with the ME49-FLuc strain were harvested, homogenized, and tissue cysts isolated on Percoll gradients, as described previously[9,27]. Following isolation, purified tissue cysts were treated with acid-pepsin solution (170 nM sodium chloride (NaCl), 60 mM hydrochloric acid (HCl) and freshly made pepsin (0.1 mg/ mL in 1xPBS) for 10 min at 37 °C. Following acid-pepsin treatment and addition of neutralization buffer (94 mM sodium carbonate (Na$_2$CO$_3$), liberated bradyzoites were equally distributed into 6-well plates with 5 mL culture media containing compounds or vehicle control (0.1–0.2% DMSO final concentration). Plates were incubated undisturbed for 12–14 days followed by fixation in 100% ethanol (ETOH) for 5 min at RT, rinsing with tap water, staining in 0.1% crystal violet solution for 10 min at RT, rinsing in water and air drying. Plaques were quantified on a Zeiss AxioObserver microscope equipped with a ×2.5 objective (Department of Molecular Microbiology Imaging Facility at the Washington University School of Medicine).

*In vivo efficacy studies.* Testing of compounds to prevent reactivation of chronic infection was based two different protocols were used to test the efficacy of PheRS inhibitors against chronic infection. In both cases, male and female C57BL/6 Ifngr1$^{-/-}$ mice were infected by oral gavage with 5 cysts of the ME49-Fluc strain isolated from chronically infected CBA/J mice. Compound BRD7929 was formulated at 1 mg/mL in freshly prepared compound resuspension buffer (vehicle; 10% ETOH, 4% Tween-80 in 1xPBS), aliquoted for single use and stored at −20 °C. In the first protocol, at 48 h post-infection, animals were given sulfadiazine (0.25 g/L in drinking water) for 20 days post-infection to prevent death during acute infection and to allow establishment of chronic infection. Two days after completion of the sulfadiazine treatment, one cohort of mice were treated once a day with 10 mg/kg BRD7929 by oral gavage for 16 days and control mice received an equal volume of vehicle during the same time frame. Animals were monitored for survival and weight loss for 30 days post-treatment. In a second protocol based on a previously published method[27], mice were treated at 48 h post-infection with sulfadiazine (sulfa; 0.25 g/L in drinking water) for 20 days or treated with 10 mg/kg BRD7929 by oral gavage once a day for 10 days or 20 days, respectively. All mice were monitored for survival and weight loss every two days for the duration of treatment and for an additional >20 days post-treatment. For bioassay, surviving mice were sacrificed, brains excised and homogenized in 1 mL of 1x PBS and 300 μL (~1/3 of donor brain) was administered to naïve C57BL/6 female Ifngr1$^{-/-}$ by oral gavage. Recipient mice were monitored for 20 days for survival, weight loss, and seroconversion by ELISA.

*Testing toxicity against host cells.* For the toxicity screen, compounds were tested in 96-well plates using all the wells as no edge affect was observed. Compounds were diluted to 40 μM (2x concentration in 10% DMEM, 0.4% DMSO) before step-wise, 3-fold dilution in 10% DMEM to create 10-dose series (20 μM to 0.001 μM). Host cells were plated in black, μ-Clear 96-well plates (Greiner Bio-One). We compared several lineages including HepG2 (human hepatocellular carcinoma (ATCC-HB-8065)), THP-1 (human monocytic tumor line), SH-SY5Y (human neuroblastoma (ATCC CRL-2266)), A549 (human lung carcinoma (ATCC CRM-CCL-185)) and Caco-2 (human intestinal adenocarcinoma (ATCC HTB-37)) and HFF (primary human foreskin fibroblast) obtained from the laboratory of John Boothroyd at Stanford University. Cells were maintained in culture media according to the formulations recommended by ATCC. Cells were plated at a density (ranging from of $5 \times 10^3$ to $2 \times 10^4$ cells/well (100 μL vol)) resulting in sub-confluent monolayers to allow expansion during the 2 day growth assay. THP-1 cells were treated with 10 ng/ml phorbol 12-myristate 13-acetate (PMA) for 24 h to differentiate into macrophages before compound addition. For all other cells, compounds were added at 6 h post seeding to host cells containing plates (200 μL final volume, 0.2% DMSO) and incubated in culture medium at 37 °C supplemented with 5% $CO_2$. At 44 h post compound addition (50 h total growth time), culture media was aspirated and replaced with Live Imaging solution (ThermoFisher) supplemented with 10 μg/mL Hoechst 33342 (Sigma) for 20 min at 37 °C. The plates were imaged on an InCell Analyzer 2000 (DAPI, ×10 objective, 4 images/well) and nuclei quantified using the InCell Developer Software package (v1.9) All media handling, compound dilution and imaging steps were directed by a Dual Pod Biomek FX driven by the SAMI EX software system to ensure accurate and reproducible data across replicates (High-Throughput Screening Center, Washington University School of Medicine). Alternatively, plates fixed with 4% formaldehyde for 10 min and stained with 10 μg/mL Hoechst 33342 (Sigma) for 20 min and imaged using a Cytation 3 multi-mode imager (DAPI, 10x objective, 4 images/well) and nuclei quantified using the Gen5 imager software (v3.08). Assays were repeated with two technical replicates within each of two independent biological replicates.

**Statistical analysis**. All statistical analyses were conducted using Prism 9 (GraphPad Software, Inc.). Dose–response inhibition curves for parasite and host cell toxicity screens ($EC_{50}$ values) were generated using (Log(inhibitor) vs. normalized response—Variable slope) function. $EC_{90}$ and $3xEC_{90}$ values were generated using (Log (agonist) vs normalized response—FindECanything) calculator. For multiple comparisons, two-way ANOVA analysis function using either Sidák's multiple comparisons test or Dunnett's multiple comparisons test was applied as indicated.

**Reporting summary**. Further information on research design is available in the Nature Research Reporting Summary linked to this article.

## Data availability
The datasets for whole genome sequencing generated during the current study are available in the short read archive (SRA) of NCBI under the accession number PRJNA731915. PDB files generated in previous studies and used here include PDB 3L4G (https://www.rcsb.org/structure/3l4g) and PDB 2IY5 (https://www.rcsb.org/structure/2IY5). All other data are found in the paper, in the supplementary information files, or source data that are provided with this paper. Unique materials described in this report are available under standard Material Transfer Agreements that can be arranged by contacting the corresponding author. Source data are provided with this paper.

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

## Acknowledgements

The authors thank Dr. Maxene Ilagan, Director of the High-Throughput Screening Core at Washington University School of Medicine, for assistance with compound and toxicity screens, Cindy S. Hon, Broad Institute, for logistical support, Mary S. Dhason for generating expression plasmids, Daniel Howe for providing the β-galactosidase *Neospora caninum* strain, Louis Weiss for providing antibodies to BAG1, Jennifer L. Barks for tissue culture support, and John M. Knapp, Broad Institute, for assistance with small molecule characterization. Supported by a grant from the NIH (AI143857 to L.D.S. and A.K.C.). P.M. is supported by the Academy of Scientific and Innovative Research (AcSIR) of India.

## Author contributions

Conceived and designed the experiments: J.B.R, M.S., P.M., B.M., E.C., S.L.S., A.S., A.K.G., A.K.C., L.D.S.; Performed the experiments: J.B.R, B.M., M.S., P.M., Y.F., T.U.; Analyzed the data: J.B.R, M.S., P.M., Y.F., T.U.; Provided critical materials A.G., B.M.; Provided critical advice and input on design of experiments: B.M., E.C., S.L.S.; Supervised the work: A.S., A.K.G., A.K.C., L.D.S.; Wrote the manuscript: J.B.R., B.M., L.D.S.; Contributed to revisions of the manuscript, all authors.

## Competing interests

The Broad Institute Inc., the President and Fellows of Harvard College, and Washington University have applied for a patent (PCT/US2019/051686 listing J.B.R., E.C., B.M. and L.D.S. as inventors) for the use of bicyclic azetidines for antiparasitic therapies. All other authors declare no competing interests.

## Additional information

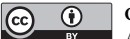

