## [Peer Review File · Nature Communications]

Bicyclic azetidines target acute and chronic stages of *Toxoplasma gondii* by inhibiting parasite phenylalanyl t-RNA synthetaseReviewers' Comments:

Reviewer #1:

Remarks to the Author:

This paper by Radke et al describes the efficacy of phenylalanine tRNA synthetase (PheRS) inhibitors in *Toxoplasma gondii*, as well as biochemical and genetic studies for validation of PheRS as the primary target. The lead compound identified in this study has very favorable pharmacokinetic properties and provided excellent protection from acute infection as well as partial protection from chronic infection in an immunocompromised mouse model of toxoplasmosis.

The paper covers many highly exciting aspects in the drug development process, the work has been carried out with great expertise, and is excellently written. The fact that this compound can target *Toxoplasma* cysts in chronically infected animals renders this paper highly valuable, and of prime interest for those interested in anti-parasitic drug development, but also for a wider audience in my opinion. I do have, however, a few remarks and some questions that could be addressed by the authors:

1. The title indicates that "multiple stages" are targeted. I would indicate that the tachyzoite and bradyzoite stage is targeted, sporozoites and merozoites are not investigated.
2. Lane 121: There seems really interesting correlation in potency between *C. parvum*, *P. falciparum* and *T. gondii*. The authors indicate that: "Furthermore, there was no significant correlation of potency for the eight bicyclic azetidines screened against *T. gondii* and *N. caninum* (Fig. S1)". I see only data for 6 compounds. In addition: the fact that there is no correlation to *N. caninum* is rather surprising given the close phylogenetic relationship of *N. caninum* and *T. gondii*. Any explanation?
3. The authors indicate that BRD3444 inhibits *C. parvum* oocyst shedding...is there a chance that it would also inhibit *T. gondii* sexual development and oocyst shedding?
4. BRD7929 inhibits PheRS, and the authors have produced resistant parasites that carry distinct mutations. I wonder whether the authors could provide any data on ultrastructural alterations in the tachyzoites upon treatments with the drug. In addition, are the resistant parasites in any way structurally different than the wild type, or are there any differences in virulence compared to the wild type?
5. Table 2 shows host toxicity values by depicting HEPG2 cell EC50s. Are there data on other cell types? I would think, since effects on tissue cysts are an important aspect of this work, probably one or more neuronal cell lines or even primary cultures should be assessed.
6. *T. gondii* infection can become problematic when primary infection takes place during pregnancy. Have the authors checked whether their drugs would in any way be safe to apply in pregnant animals?
7. Mice are orally infected with infected brains containing tissue cysts. Are there data available on protective effects when mice are infected with oocysts? This would surely be an important infection route to investigate
8. The claim that BRD7929 is active against *T. gondii* bradyzoites is based on a very artificial mouse model. While I explicitly do not question the validity of the results that are shown here, I believe a better way to assess for activity against bradyzoites would be to infect mice under conditions where they develop tissue cysts, and then actually treat those mice with the compound and assess the effects on the tissue cysts, either by assessing their infectivity, or possibly by molecular detection of viability markers and preferentially also visual methods (histology). I can see that this is a lot of work, but this would clearly provide a clear picture on the claim that these compounds could be useful for treatment of chronic infection.

Reviewer #2:

Remarks to the Author:

This manuscript describes the discovery of a series of bicyclic azetidines active against different stages against different stages of *Toxoplasma gondii*, by inhibiting cytosolic phenylalanine tRNA synthase. Especially the activity found against ex vivo bradyzoites and the favorable PK profile for CNS drug

exposure are promising with respect to translating the current findings for the treatment of chronic toxoplasmosis.

The work is important and carefully executed. Yet, I feel the findings are not innovative enough to warrant publication in Nature Communications, given that these or very similar compounds have already been reported to confer activity against evolutionary related parasites, i.e. *P. falciparum* and *C. parvum*, via inhibition of the PheRS, here also validated as the primary target of the bicyclic azetidines in *T. gondii*.

Reviewer #3:

Remarks to the Author:

Summary

In this study, Radke et al. investigated the potential effect of bicyclic azetidines, currently in development for malaria, and proven potent against cryptosporidiosis, against *Toxoplasma gondii*. Authors elegantly designed the study and provided a comprehensive and well-written manuscript with excellent results extending the efficacy of these compounds against another apicomplexan and highly prevalent parasite, *T. gondii*. The study was performed on tachyzoites in vitro, on ex-vivo isolated bradyzoites and an in vivo susceptible murine model to *T. gondii*. Moreover, authors demonstrated that the target of bicyclic azetidines was TgPheRS, following mutations of key residues and after testing the lead compound against the human versus Tg recombinant protein. Toxicity on HepG2 cells, PK and bioavailability studies were also provided adding to the strength of the presented results. The below points need to be addressed to further improve the study.

- Authors showed the effect of bicyclic azetidines on tachyzoites in vitro, and bradyzoites in vitro and in murine models. Modify the title to better reflect the findings in the tested systems and tune-down "multiple-life cycle stages".
- Lane 38 in the abstract: "currently approved drugs are not capable of clearing chronic infection". This statement is not accurate unless authors mention human toxoplasmosis. A recently published paper proved the efficacy of Imiquimod, which is FDA approved, against acute and more importantly chronic toxoplasmosis in murine models. Modify the abstract and cite this paper in the discussion.
- In figure 1A, authors tested the effect of the 28 PheRS compounds against *T.gondii* tachyzoites in vitro, using the luciferase based tachyzoite growth assay in confluent HFF monolayers. The results are very convincing. Yet, even if HFF which are commonly and world widely used to support *T. gondii* growth, these cells are not the appropriate host cells hijacked by the parasite, the assay should be performed in more appropriate host cells such as macrophages (whether cells lines or primary macrophages), at least for the 4 compounds BRD3444, BRD8494, BRD7929 and BRD2987, to strengthen the results.
- If BRD3444 was one of the most potent inhibitors of *T.gondii* and mutations conferring resistance to BRD3444 are conserved between *Plasmodium falciparum* and *Cryptosporidium parvum*, what is the rationale of generating resistant *T.gondii* to BRD7929 and not BRD3444? It is not well introduced why did the authors use the 4 most potent PheRS inhibitors, to study correlation of growth inhibition between the three apicomplexan parasites and then select eight and screen the EC50 against *Neospora caninum*, then choose BRD7929 for *T. gondii* resistance studies.
- In figure 3, authors generated 2 resistant independent pools of RH-Fluc to BRD7929. First of all, why did the authors choose a type I strain which doesn't form cysts in vivo (especially that they tested different genotypes in table 1), for resistance studies. One can expect these studies in a type II strain, particularly that they screened later for the efficacy of these inhibitors on bradyzoite stages. Any difference between the sequence of PheRS between tachyzoites and bradyzoites? It would be interesting to generate a resistant type II strain and check if similar results are obtained.
- In the ex-vivo treated bradyzoite assay, why did the authors choose to pursue their studies of EC90 and 3xEC90 for BRD7929 and BRD8494? Any rationale behind this choice? Did the BRD3444 and 2987 give same results? Did the authors assess constant treatment for less than 12-14 days to observe what is the minimal required time for these inhibitors to exhibit activity?

- Did the authors assess the activity of these inhibitors on cultured bradyzoites after in vitro switch?
- In the pharmacokinetics studies, authors chose again the BRD7929. From a reader perspective, one cannot easily follow the choice of 8 in some sections, 4 in other sections, one in resistance studies, 4 then 2 in ex-vivo treated bradyzoite assays then one in pharmacokinetics and in vivo studies. Although the study is overall nicely designed, but jumping a such between the inhibitors is misleading and requires a justification of the choice of inhibitor(s) in each section.
- How did the authors choose the dose of 10mg/kg in vivo for both PK studies and treatment regimens? Was that based on previous studies or did the authors test different concentrations in vivo and picked up this dose?
- Since BRD7929 proved efficient against tachyzoites in vitro, did the authors test for its potency against acute toxoplasmosis in vivo?
- Lane 215, the term immunocompromised mice is misleading and one can suspect that authors tested for the activity of BRD7929 in SCID or NSG mice following oral gavage with cysts of *T. gondii*. Even if the DOS libraries allowed the identification of the tested inhibitors, in an IFN-g synergistic fashion, explaining the choice of IFN-g KO mice which are susceptible for the infection, did the authors treat SCID or NSG mice with BRD7929, following oral gavage with cysts from ME49-Fluc strain? What is the number of cysts that was used for oral gavage (not mentioned in the methods section)?
- To study the potential of BRD7929 as treatment for reactivation, did the authors treat chronically infected wild-type mice (not IFN gamma KO mice) and then assess reactivation following immunosuppression (for example after dexamethasone treatment)?
- Show the ELISA and Bioassay results as supplementary figures to prove that death occurred due to the parasite and not due to other factors.

Minor comments

- Italicize "in vivo and in vitro" all over the text.
- Unify the tense in the abstract. The present tense would be preferable.
- Lanes 75 and 86, add the references supporting the effect of Bicyclic azetidines on PheRS in apicomplexa.

REVIEWER COMMENTS

Reviewer #1 (Remarks to the Author):

This paper by Radke et al describes the efficacy of phenylalanine tRNA synthetase (PheRS) inhibitors in *Toxoplasma gondii*, as well as biochemical and genetic studies for validation of PheRS as the primary target. The lead compound identified in this study has very favorable pharmacokinetic properties and provided excellent protection from acute infection as well as partial protection from chronic infection in an immunocompromised mouse model of toxoplasmosis.

The paper covers many highly exciting aspects in the drug development process, the work has been carried out with great expertise, and is excellently written. The fact that this compound can target *Toxoplasma* cysts in chronically infected animals renders this paper highly valuable, and of prime interest for those interested in anti-parasitic drug development, but also for a wider audience in my opinion. I do have, however, a few remarks and some questions that could be addressed by the authors:

1. The title indicates that “multiple stages” are targeted. I would indicate that the tachyzoite and bradyzoite stage is targeted, sporozoites and merozoites are not investigated.

We have changed the title to: “Bicyclic azetidines targets acute and chronic stages of *Toxoplasma gondii* by inhibiting parasite phenylalanyl t-RNA synthetase” to better reflect the data provided in our study.

2. Lane 121: There seems really interesting correlation in potency between *C. parvum*, *P. falciparum* and *T. gondii*. The authors indicate that: “Furthermore, there was no significant correlation of potency for the eight bicyclic azetidines screened against *T. gondii* and *N. caninum* (Fig. S1)”. I see only data for 6 compounds. In addition: the fact that there is no correlation to *N. caninum* is rather surprising given the close phylogenetic relationship of *N. caninum* and *T. gondii*. Any explanation?

We thank the reviewer for pointing out this inconsistency. We have re-examined the original data and found that the low r^2 value was driven by a single compound BRD7929 that was much less potent on *N. caninum* than *T. gondii*. If we exclude this data point, the r^2 was in fact quite good (> 0.6). Because we suspected the original data may have been erroneous (perhaps due to an unstable stock or dilution error?), we repeated the assay to measure the effect of BRD7929 on *N. caninum* growth in three biological replicates each with two technical replicates. The new value that we determined for the EC_{50} of BRD7929 is 0.042 nM (Table S3), which is much more consistent with the potent activities of other compounds on *N. caninum* (Fig. S1). As a result of correcting this data, the new r^2 for the linear regression comparing activities against *T. gondii* and *N. caninum* is now 0.69, which is statistically significant ($P < 0.05$). We revised the text to indicate that there is in fact a significant correlation in the growth inhibitory activities between *T. gondii* and *N. caninum*, in keeping with the conserved activity of PheRS inhibitors across the Apicomplexa.

3. The authors indicate that BRD3444 inhibits *C. parvum* oocyst shedding...is there a chance that it would also inhibit *T. gondii* sexual development and oocyst shedding?

This is an interesting idea for future study. Unfortunately, we are currently not able to test infections in cats as these studies have been banned at the USDA where we formerly conducted such trials. We have acknowledged in the Discussion that it would be of interest to test the effects of inhibitors against other life cycle stages.

4. BRD7929 inhibits PheRS, and the authors have produced resistant parasites that carry distinct mutations. I wonder whether the authors could provide any data on ultrastructural alterations in the tachyzoites upon treatments with the drug. In addition, are the resistant

parasites in any way structurally different than the wild type, or are there any differences in virulence compared to the wild type?

We have not examined the ultrastructure of parasites treated by the compounds used here, but feel these data would only be descriptive and likely would not reveal differences that would inform us about the role of PheRs or the mechanism of resistance in the mutants. Instead, we have tested them in a lytic growth assay and by plaquing on monolayers of host cells. These assays capture the ability of mutants to invade, replicate, egress and spread in the absence of compounds. We did not detect any growth impairment that would suggest that these mutations might impart fitness defects. These new data are found in Supplemental Fig. S3. We have revised the text to include a summary of these findings in the results.

5. Table 2 shows host toxicity values by depicting HEPG2 cell EC50s. Are there data on other cell types? I would think, since effects on tissue cysts are an important aspect of this work, probably one or more neuronal cell lines or even primary cultures should be assessed.

This is an excellent suggestion and we have now tested the potency of the 4 top compounds on multiple cell types of different lineages. Similar to the HepG2 cells, the compounds showed minimal activity against human epithelial (Caco-2), endothelial (A549) and fibroblast (HFF) lineages (Table 2). However, we did observe higher levels of growth inhibition against monocytic (THP-1) and neuronal (SH-SY5Y) cells as shown in Table 2. Despite this increased sensitivity, the compounds still demonstrate > 10 fold selectivity (in most cases > 30 fold) in inhibiting parasite growth vs. host cell growth. We have acknowledged this limitation in the Discussion as a challenge that future medicinal chemistry efforts will need to address.

6. *T. gondii* infection can become problematic when primary infection takes place during pregnancy. Have the authors checked whether their drugs would in any way be safe to apply in pregnant animals?

We have not tested models of congenital infection but at this point we feel this would be premature given the toxicity issues we encountered with the lead molecule. We look forward to testing compounds in pregnancy models once we are able to identify appropriate compounds.

7. Mice are orally infected with infected brains containing tissue cysts. Are there data available on protective effects when mice are infected with oocysts? This would surely be an important infection route to investigate.

This is an interesting suggestion and something we look forward to testing in the future. We have added a statement to the Discussion to indicate that further testing against different infectious forms, and in different models, including pregnancy would be appropriate once suitable leads are identified.

8. The claim that BRD7929 is active against *T. gondii* bradyzoites is based on a very artificial mouse model. While I explicitly do not question the validity of the results that are shown here, I believe a better way to assess for activity against bradyzoites would be to infect mice under conditions where they develop tissue cysts, and then actually treat those mice with the compound and assess the effects on the tissue cysts, either by assessing their infectivity, or possibly by molecular detection of viability markers and preferentially also visual methods (histology). I can see that this is a lot of work, but this would clearly provide a clear picture on the claim that these compounds could be useful for treatment of chronic infection.

Although the suggested model for monitoring chronic infection is widely used, it lacks precision and does not provide a robust endpoint. Typically investigators look for a reduction in cyst number. However, it is not possible to assure that cysts are completely cleared due to the inherent insensitivity of the assay. Treatments to render mice susceptible (using neutralizing antibodies to IFN γ or dexamethasone) are only partially effective at revealing such chronic infections. Hence, the best outcome we might expect from this experiment is a reduction in cysts, perhaps below the threshold of detection – but without definitive evidence for cure. It is

also unclear that cyst reduction has a clinical benefit since if any cysts remain, immunocompromised individuals would still be at risk. For this reason, we prefer the reactivation model using immunocompromised animals because if any residual tachyzoites or bradyzoites remain, the infection will reactivate and the animal will die. In contrast, animals that survive in this model provide a very robust end point for cure.

To support the conclusions from our in vivo studies, we have extended the studies of in vitro derived bradyzoites to show that the compounds have excellent activity against early stages of bradyzoite conversion as well as mature in vitro bradyzoites (Fig. 5 A-F). Combined with the study showing compounds have activity on ex vivo bradyzoites (Fig. 5 G-J), these studies demonstrate direct activity of PheRS inhibitors on bradyzoites. In summary, we feel that the current studies are sufficient to provide proof of principle that PheRS inhibitors are effective against both acute and chronic stages and thus can delay or prevent reactivation in an immunocompromised animal. We have added a statement in the Discussion to acknowledge important results obtained from other models and to emphasize why we prefer the reactivation model used here.

Reviewer #2 (Remarks to the Author):

This manuscript describes the discovery of a series of bicyclic azetidines active against different stages against different stages of *Toxoplasma gondii*, by inhibiting cytosolic phenylalanine tRNA synthase. Especially the activity found against ex vivo bradyzoites and the favorable PK profile for CNS drug exposure are promising with respect to translating the current findings for the treatment of chronic toxoplasmosis.

The work is important and carefully executed. Yet, I feel the findings are not innovative enough to warrant publication in Nature Communications, given that these or very similar compounds have already been reported to confer activity against evolutionary related parasites, i.e. *P. falciparum* and *C. parvum*, via inhibition of the PheRS, here also validated as the primary target of the bicyclic azetidines in *T. gondii*.

We can appreciate the perspective of this reviewer, but would also like to point out that activity against one apicomplexan does not always lead to activity against all. For example, we recently published a summary of the potency of anti-malarial drugs – either clinically approved or late stage development - and found a very poor correlation with activity against *T. gondii*. Included in this list are PI4K and ATPase4 inhibitors that show promise for *P. falciparum* and *C. parvum* but have little activity on *T. gondii*. As well, the combination of pyrimethamine-sulfa that works on *T. gondii* and *P. falciparum* does not work on *C. parvum* (due to salvage of thymidine). Hence, the conserved nature of the PheRS target across the Apicomplexa is highly unusual and makes our report noteworthy. We have added a statement to the Discussion to emphasize this point.

Reviewer #3 (Remarks to the Author):

Summary

In this study, Radke et al. investigated the potential effect of bicyclic azetidines, currently in development for malaria, and proven potent against cryptosporidiosis, against *Toxoplasma gondii*. Authors elegantly designed the study and provided a comprehensive and well-written manuscript with excellent results extending the efficacy of these compounds against another apicomplexan and highly prevalent parasite, *T. gondii*. The study was performed on tachyzoites in vitro, on ex-vivo isolated bradyzoites and an in vivo susceptible murine model to *T. gondii*. Moreover, authors demonstrated that the target of bicyclic azetidines was TgPheRS, following mutations of key residues and after testing the lead compound against the human versus Tg recombinant protein. Toxicity on HepG2 cells, PK and bioavailability studies were also provided adding to the strength of the presented results. The below points need to be addressed to further improve the study.

- Authors showed the effect of bicyclic azetidines on tachyzoites in vitro, and bradyzoites in vitro and in murine models. Modify the title to better reflect the findings in the tested systems and tune-down "multiple-life cycle stages".

We have changed the title to: "Bicyclic azetidines targets acute and chronic stages of *Toxoplasma gondii* by inhibiting parasite phenylalanyl t-RNA synthetase" to better reflect the data provided in our study.

- Lane 38 in the abstract: "currently approved drugs are not capable of clearing chronic infection". This statement is not accurate unless authors mention human toxoplasmosis. A recently published paper proved the efficacy of Imiquimod, which is FDA approved, against acute and more importantly chronic toxoplasmosis in murine models. Modify the abstract and cite this paper in the discussion.

Thank you for pointing out this distinction. We have changed the statement in the abstract to: *currently approved drugs are not capable of clearing chronic infection in humans*. We also appreciate the reviewer pointing out the reference that we agree provides important new data. We have cited this reference in the discussion of other compounds that are effective in reducing cyst burden, although none of these have been shown to cure chronic infection in immunocompromised animals.

- In figure 1A, authors tested the effect of the 28 PheRS compounds against *T. gondii* tachyzoites in vitro, using the luciferase based tachyzoite growth assay in confluent HFF monolayers. The results are very convincing. Yet, even if HFF which are commonly and world widely used to support *T. gondii* growth, these cells are not the appropriate host cells hijacked by the parasite, the assay should be performed in more appropriate host cells such as macrophages (whether cells lines or primary macrophages), at least for the 4 compounds BRD3444, BRD8494, BRD7929 and BRD2987, to strengthen the results.

Toxoplasma infects all nucleated cell types including both hematopoietic and non-hematopoietic cells and so HFF cells are a highly relevant cell type. To extend our studies, we have also tested the potency of the 4 top compounds on multiple cell types of different lineages. Similar to the HepG2 cells, the compounds showed minimal activity against human epithelial (Caco-2), endothelial (A549) and fibroblast (HFF) lineages (Table 2). However, we did observe higher levels of growth inhibition against monocytic (THP-1) and neuronal (SH-SY5Y) cells as shown in Table 2. Despite this increased sensitivity, the compounds still demonstrate > 10 fold selectivity (in most cases > 30 fold) in inhibiting parasite growth vs. host cell growth. We have acknowledged this limitation in the Discussion as a challenge that future medicinal chemistry efforts will need to address.

- If BRD3444 was one of the most potent inhibitors of *T.gondii* and mutations conferring resistance to BRD3444 are conserved between *Plasmodium falciparum* and *Cryptosporidium parvum*, what is the rationale of generating resistant *T.gondii* to BRD7929 and not BRD3444? It is not well introduced why did the authors use the 4 most potent PheRS inhibitors, to study correlation of growth inhibition between the three apicomplexan parasites and then select eight and screen the EC50 against *Neospora caninum*, then choose BRD7929 for *T. gondii* resistance studies.

The rationale for choice of various analogs was based not only on prior studies, but the potency against *T. gondii*, especially in the ex vivo bradyzoite killing assay. BRD7929 is clearly superior in this assay and has better PK properties for in vivo studies. So for the generation of resistant mutants, we chose to use BRD7929 due to its potency and better PK properties that made it more suitable for follow up studies in vivo. The choice of BRD3444 in the comparison assays with *Plasmodium* was driven largely by the fact that we had all eight stereoisomers available for testing in parallel. The conformers are shared among different analogs, so the information derived from this compound is likely relevant to others in the series. The choice of compounds for comparison against *T. gondii* and *N. caninum* was based both on prior data with *P. falciparum* and *C. parvum* as well as our results with *T. gondii*. We have modified the text to more fully explain the rationale for choice of compounds at each step.

- In figure 3, authors generated 2 resistant independent pools of RH-Fluc to BRD7929. First of all, why did the authors choose a type I strain which doesn't form cysts in vivo (especially that they tested different genotypes in table 1), for resistance studies. One can expect these studies in a type II strain, particularly that they screened later for the efficacy of these inhibitors on bradyzoite stages. Any difference between the sequence of PheRS between tachyzoites and bradyzoites? It would be interesting to generate a resistant type II strain and check if similar results are obtained.

We chose the type I strain as it is the most commonly used genotype for laboratory studies. For example, it is much easier to engineer point mutants into this background due to available genetic tools. Indeed, it would also be interesting to perform these selection studies in a type II strain, although we do not think the outcome would differ. There are no differences in the gene expressed by tachyzoites and bradyzoites nor are there any polymorphisms between strains. As well, we show that the compounds have near equal potency against multiple different genetic lineages. As such, we feel the data from the type I RH strain is representative of what would occur with other strains. We have added a statement to the Discussion to clarify this point.

- In the ex-vivo treated bradyzoite assay, why did the authors choose to pursue their studies of EC90 and 3xEC90 for BRD7929 and BRD8494? Any rationale behind this choice? Did the BRD3444 and 2987 give same results? Did the authors assess constant treatment for less than 12-14 days to observe what is the minimal required time for these inhibitors to exhibit activity?

We chose these levels because they allow us to compare the potency of PheRS inhibitors to prior compounds that we have worked with. The choice is somewhat arbitrary but it has advantages over simply using EC₅₀ since the slope of the inhibition curve and hence EC₉₀ can vary appreciably. We also typically design in vivo studies to monitor the same cutoffs based on PK studies. We agree the choice is somewhat arbitrary, but it nonetheless provides useful comparisons. We have not defined the minimal time nor minimal conc. to achieve complete kill and the choice of 14 days treatment was based on the time needed for plaques to develop in the control. We did not test the other analogs as the comparison between BRD7929 and BRD8494 already presented a range of different outcomes and BRD7929 was best suited for follow up in vivo studies, as described above. We anticipate doing further studies to define the

minimum conc. and time needed for irreversible treatment once more suitable analogs are available.

- Did the authors assess the activity of these inhibitors on cultured bradyzoites after in vitro switch?

This is an excellent suggestion and we have added new data to address this point in Fig. 5. We have developed two assays to monitor activity against in vitro differentiated bradyzoites. Initially, we used early stages of differentiation (3 days) followed by treatment with bicyclic azetidines inhibitors for 2 days to demonstrate that they block further growth of cysts (Fig. 5 A-C). Additionally, we have used longer time frames of differentiation (14 days) to demonstrate that bicyclic azetidines inhibitors administered for 48 hr after 12 days of induction are able to prevent the outgrowth of bradyzoites in a subsequent plaque assays (Fig. 5 D-F). In combination with the demonstrated activity against ex vivo purified bradyzoites (Fig. 5 G-J), these findings demonstrate convincing activity against the chronic stages of *T. gondii*.

- In the pharmacokinetics studies, authors chose again the BRD7929. From a reader perspective, one cannot easily follow the choice of 8 in some sections, 4 in other sections, one in resistance studies, 4 then 2 in ex-vivo treated bradyzoite assays then one in pharmacokinetics and in vivo studies. Although the study is overall nicely designed, but jumping a such between the inhibitors is misleading and requires a justification of the choice of inhibitor(s) in each section.

The choices were dictated by potency on tachyzoites as well as known PK properties. BRD7929 is much more potent against bradyzoites in the ex vivo assay and it has a much longer half live in mice, so we reasoned it would provide greater exposure. We have modified the text to make the rationale for these choices clear.

- How did the authors choose the dose of 10mg/kg in vivo for both PK studies and treatment regimens? Was that based on previous studies or did the authors test different concentrations in vivo and picked up this dose?

The choice was based on the properties of this compound in standard PK studies (10 mg /kg is the most commonly employed dose for initial PK in mouse) as well as prior experience with *Plasmodium*. We have modified the text to make the rationale for this choice clear.

- Since BRD7929 proved efficient against tachyzoites in vitro, did the authors test for its potency against acute toxoplasmosis in vivo?

The assay we are using essentially captures both acute and chronic stages. In the experiment shown in Fig.6 B,C, the compounds are administered during the acute phase and they must inhibit tachyzoite growth sufficiently to prevent death as well as block bradyzoites that would generate chronic infection. In the reactivation model shown in Fig. 6 D,E, when sulfadiazine is removed in this immunocompromised model, cyst rupture leads to reversion to tachyzoites and unchecked growth. The PheRS inhibitors need to kill bradyzoites and/or block tachyzoite re-emergence in order to protect the mice. We prefer this model as it provides a combined readout while using fewer animals. We have revised the Discussion to make this point clear.

- Lane 215, the term immunocompromised mice is misleading and one can suspect that authors tested for the activity of BRD7929 in SCID or NSG mice following oral gavage with cysts of *T. gondii*. Even if the DOS libraries allowed the identification of the tested inhibitors, in an IFN-g synergistic fashion, explaining the choice of IFN-g KO mice which are susceptible for the infection, did the authors treat SCID or NSG mice with BRD7929, following oral gavage with cysts from ME49-Fluc strain? What is the number of cysts that was used for oral gavage (not mentioned in the methods section)?

Immunocompromised mice come in a wide variety of genotypes with different deficiencies and *lfngr^{-/-}* is just one example. We prefer this model as the animals are fully susceptible and cannot survive infection at any dose of viable parasites without chemotherapy. The other mice

suggested here (SCID or NSG) are alternative immunocompromised models, albeit they are less impaired as they retain some components of innate immunity. We agree these would be interesting models for future studies. We have revised the methods to include the dose of cysts used for challenge.

- To study the potential of BRD7929 as treatment for reactivation, did the authors treat chronically infected wild-type mice (not IFN gamma KO mice) and then assess reactivation following immunosuppression (for example after dexamethasone treatment)?

In our experience, this model is less precise as it only leads to a partially immunocompromised state. Hence, some animals may remain infected and not reactivate, making it very difficult to provide a precise endpoint for treatment. This is the reason we prefer the immunocompromised model used here.

- Show the ELISA and Bioassay results as supplementary figures to prove that death occurred due to the parasite and not due to other factors.

The dose of parasites used here was chosen based on numerous previous experiments that were used to define the kinetics of infection. Thus we are 100% certain that deaths are attributable to parasite infection and not due to other causes. Control animals in our colony experienced no signs of illness or death during these trials. For the data presented in Fig. 6C we have included ELISA data for animals that survived at day 30 (see Fig. S3). The surviving 5 animals were clearly seropositive, indicating they were infected (the antigen load from the inoculum is far too low to produce this effect) (Fig. S3). However, none of these animals were chronically infected as confirmed by bioassay. We conclude from these results that the treatment was curative. We have also included the weight and ELISA results for recipient animals in the bioassay experiment confirming that these animals were not infected by the inoculation of tissues from surviving animals (Fig. S3). We have also included the ELISA value for the single animal in the trial shown in Fig. 6E, again confirming that it was infected yet it failed to succumb to infection, indicating it was cured by treatment (Fig. S3). We have updated the text to better explain the assay and outcome of these trials.

Minor comments

- Italicize “in vivo and in vitro” all over the text.

The format is left to the discretion of the journal as publishers vary considerably on the use of italics.

- Unify the tense in the abstract. The present tense would be preferable.

We have used present tense for published work and past tense for new observations in keeping with the guidelines for the Council of Scientific Editors. If the journal prefers a different format, we are happy to accommodate.

- Lanes 75 and 86, add the references supporting the effect of Bicyclic azetidines on PheRS in apicomplexa.

Revised as suggested.

Other data not requested

We have added IC₅₀ values for the single and one of the double mutants to the enzyme inhibition data shown in Fig. 4B. As well, the IC₅₀ and EC₅₀ values are compiled in Table S4 and a lineage regression analysis ($r^2 = 0.9$) is provided to show that the growth inhibition and enzyme inhibition are tightly correlated.

Reviewers' Comments:

Reviewer #1:

Remarks to the Author:

I have read the responses of the authors to my concerns, and also those of the other reviewers, and in my opinion all crucial points were addressed in a satisfactory way. I have no more issues with this paper, and believe that it should be published as it is.

Reviewer #3:

Remarks to the Author:

We thank the authors for addressing all the requested points.
The revised version is now suitable for publication.

Reviewer #1 (Remarks to the Author):

I have read the responses of the authors to my concerns, and also those of the other reviewers, and in my opinion all crucial points were addressed in a satisfactory way. I have no more issues with this paper, and believe that it should be published as it is.

We thank the reviewer for their helpful comments and recommendation for publication.

Reviewer #3 (Remarks to the Author):

We thank the authors for addressing all the requested points.
The revised version is now suitable for publication.

We thank the reviewer for their helpful comments and recommendation for publication.